# Cumulative Reasoning with Large Language Models

**Yifan Zhang**[*]
*IIIS, Tsinghua University*

*yifanzhangresearch@gmail.com*

**Jingqin Yang**[*]
*IIIS, Tsinghua University*

*yangjq21@mails.tsinghua.edu.cn*

**Yang Yuan**[†]
*IIIS, Tsinghua University*
*Shanghai Qi Zhi Institute*

*yuanyang@tsinghua.edu.cn*

**Andrew C Yao**[†]
*IIIS, Tsinghua University*
*Shanghai Qi Zhi Institute*

*andrewcyao@tsinghua.edu.cn*

**Reviewed on OpenReview:** *https://openreview.net/forum?id=grW15p4eq2*

## Abstract

Recent advancements in large language models (LLMs) have shown remarkable progress, yet their ability to solve complex problems remains limited. In this work, we introduce Cumulative Reasoning (CR), a structured framework that enhances LLM problem-solving by emulating human-like iterative and cumulative thought processes. CR orchestrates LLMs in three distinct roles—Proposer, Verifier(s), and Reporter—to systematically decompose tasks, generate and validate intermediate reasoning steps, and compose them into a solution by building a dynamic Directed Acyclic Graph (DAG) of verified propositions. This approach substantially enhances problem-solving capabilities. We demonstrate CR's advantage through several complex reasoning tasks: it outperforms existing methods in logical inference tasks with up to a 9.3% improvement, achieving 98.04% accuracy on the curated FOLIO wiki dataset. In the Game of 24, it achieves 98% accuracy, marking a 24% improvement over previous methods. In solving MATH problems, CR achieves a 4.2% increase from previous methods and a 43% relative improvement in the most challenging level 5 problems. When incorporating a code environment with CR, we further harness LLMs' reasoning capabilities and outperform the Program of Thought (PoT) method by 38.8%. The code is available at https://github.com/iiis-ai/cumulative-reasoning.

## 1 Introduction

Large language models (LLMs) have exhibited remarkable progress across a wide range of applications (Devlin et al., 2018; Radford et al., 2018; 2019; Brown et al., 2020; Raffel et al., 2020; OpenAI, 2023). Despite these advancements, LLMs continue to encounter significant challenges when tasked with solving problems that require intricate, multi-step reasoning and robust logical inference. For example, empirical studies have shown that LLMs often struggle to generate correct solutions for high school mathematics problems (Lightman et al., 2023), highlighting a persistent gap between intuitive language generation and rigorous problem-solving.

Inspired by Kahneman's dual-process theory (Kahneman, 2011), which distinguishes between rapid, intuitive processing (System 1) and slower, deliberative reasoning (System 2), it becomes evident that current LLMs

---

[*]Equal contribution
[†]Corresponding authors

primarily operate in a System 1 mode. This limitation restricts their capacity to engage in the systematic, stepwise reasoning necessary for complex tasks.

Recent developments such as Chain-of-Thought (CoT) prompting (Wei et al., 2022) and Tree-of-Thought (ToT) methodologies (Yao et al., 2023; Long, 2023) have made significant strides by guiding LLMs through sequential and hierarchical reasoning processes. However, these approaches often lack robust mechanisms for dynamically storing, verifying, and cumulatively leveraging all validated intermediate results in a flexible manner—a critical aspect of human cognition that enables error-checking, iterative refinement, and the construction of complex arguments.

In this work, we introduce Cumulative Reasoning (CR), a reasoning method that characterizes a more holistic representation of the thinking process. CR orchestrates a symphony of three LLM roles—the proposer, verifier(s), and reporter—to iteratively propose, validate, and compile reasoning steps into a comprehensive solution. This decomposition and composition strategy effectively transforms complex, multifaceted problems into a series of manageable tasks, substantially enhancing the problem-solving capabilities of LLMs. CR's contributions include its structured, synergistic orchestration of these roles and its dynamic construction and utilization of a Directed Acyclic Graph (DAG) of validated reasoning steps. This cumulative DAG allows CR to build upon a growing, verified knowledge base specific to the problem instance, facilitating more flexible, robust, and complex reasoning pathways. Our evaluation spans three distinct areas:

1. Logical inference tasks: our method demonstrates superior performance on datasets like FOLIO wiki and AutoTNLI, with improvements of up to 9.3% and an outstanding 98.04% accuracy on a curated version of the FOLIO dataset.
2. The Game of 24: we achieved 98% accuracy, marking a 24% improvement over the existing state-of-the-art method ToT (Yao et al., 2023) while using only about 25% visited states.
3. Solving MATH problems: our method establishes new benchmarks with a margin of 4.2% over previous methods (Fu et al., 2022; Zheng et al., 2023) without external tools. Noteworthy, our method achieves notable 43% relative improvements on the hardest level 5 problems ($22.4\% \rightarrow 32.1\%$). Moreover, by integrating CR with a Python code environment—absent external aids like retrieval systems, we achieve a 72.2% accuracy on the MATH dataset, outperforming previous methods such as PoT (Chen et al., 2022) and PAL (Gao et al., 2023) with 38.8% relative improvement and demonstrating the adaptability and robustness of CR across various complex tasks.

## 2 Background

In this section, we review the formal foundations of logic that underpin our approach and present an illustrative example adapted from the FOLIO dataset (Han et al., 2022).

### 2.1 Logic

Propositional logic is the most basic formal system in logic. It is built from atomic propositions (e.g., $p$, $q$, $r$) and logical connectives such as conjunction ($\wedge$), disjunction ($\vee$), implication ($\Rightarrow$), and negation ($\neg$). In this setting, the truth values are denoted by the constants 1 (true) and 0 (false). Fundamental laws of propositional logic include:

$$x \wedge x = x, \quad x \vee x = x, \quad 1 \wedge x = x, \quad 0 \vee x = x,$$

along with the absorption law:

$$x \wedge (y \vee x) = x = (x \wedge y) \vee x,$$

and the distributive laws:

$$x \wedge (y \vee z) = (x \wedge y) \vee (x \wedge z), \quad x \vee (y \wedge z) = (x \vee y) \wedge (x \vee z).$$

In any Boolean algebra, every element $x$ has a complement $\neg x$, which satisfies:

$$x \wedge \neg x = 0, \quad x \vee \neg x = 1, \quad \neg\neg x = x.$$

Extending this framework, *first-order logic (FOL)* introduces quantifiers to reason about collections of objects. Universal quantification ($\forall$) and existential quantification ($\exists$) allow statements such as

$$\forall x \left( \text{Dog}(x) \Rightarrow \text{Animal}(x) \right),$$

which reads as "for every $x$, if $x$ is a dog, then $x$ is an animal." In contrast, *higher-order logic (HOL)* permits quantification over functions and predicates, greatly increasing expressiveness. For a detailed treatment of HOL, please refer to Appendix C.1.

## 2.2 Illustrative Example

To illustrate these concepts, consider an example adapted from the FOLIO dataset, where only natural language statements (without explicit logical formulas) are provided as context. The premises are as follows:

1. All monkeys are mammals: $\forall x \left( \text{Monkey}(x) \Rightarrow \text{Mammal}(x) \right)$.
2. Every animal is either a monkey or a bird: $\forall x \left( \text{Animal}(x) \Rightarrow (\text{Monkey}(x) \vee \text{Bird}(x)) \right)$.
3. All birds can fly: $\forall x \left( \text{Bird}(x) \Rightarrow \text{Fly}(x) \right)$.
4. Anything that can fly has wings: $\forall x \left( \text{Fly}(x) \Rightarrow \text{Wings}(x) \right)$.
5. Rock is not a mammal but is an animal: $\neg\text{Mammal}(\text{Rock}) \wedge \text{Animal}(\text{Rock})$.

The question is: *Does Rock have wings?*

A rigorous derivation proceeds as follows:

a. The contrapositive of (1) gives:

$$\forall x \left( \neg\text{Mammal}(x) \Rightarrow \neg\text{Monkey}(x) \right).$$

b. Combining (a) with (5) implies that Rock is not a monkey while still being an animal.

c. Premise (2) then entails that Rock must be either a monkey or a bird.

d. Since Rock is not a monkey (by step (b)), it follows that Rock is a bird.

e. From (3) and the conclusion that Rock is a bird, we deduce that Rock can fly.

f. Finally, applying (4) to the fact that Rock can fly, we conclude that Rock has wings.

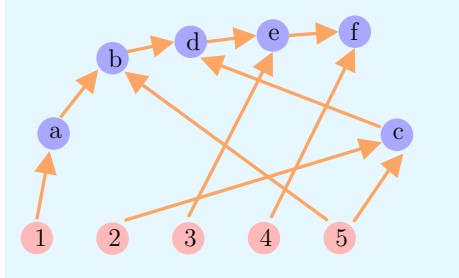

Figure 1: An illustration of the logical derivation process.

Although the derivation appears as a linear sequence of steps from (a) to (f), its underlying structure is more naturally modeled as a directed acyclic graph (DAG), where each edge represents an individual inference. This DAG structure better captures the cumulative and interdependent nature of the reasoning process.

## 3 Cumulative Reasoning

In this section, we introduce our method, Cumulative Reasoning (CR), a structured framework that leverages a collaborative process among specialized Large Language Models (LLMs) to address complex problem-solving tasks. Unlike conventional methods that generate a linear chain of thought, CR decomposes a problem into manageable sub-tasks and incrementally builds a solution by cumulatively accumulating and verifying intermediate reasoning steps.

This approach is inspired by human cognitive processes that involve iterative refinement and building upon established knowledge, as well as principles from intuitionistic logic and mathematical constructivism which emphasize the constructive nature of proofs built from validated steps (Troelstra, 1973). CR aims to operationalizes these principles for LLM-based reasoning.

CR orchestrates three distinct roles:

1. **Proposer:** Generates candidate reasoning steps based on the current context, thereby initiating each cycle of reasoning.
2. **Verifier(s):** Critically assess and validate the proposer's suggestions. The Verifier's conceptual role is to ensure logical soundness. In practice, this can be implemented by another LLM instance (acting as a self-critique mechanism) or, ideally, by more formal methods such as symbolic reasoning systems (e.g., a theorem prover) or an integrated code environment (e.g., a Python interpreter for mathematical or arithmetic validation). Only verified steps are added to the DAG.
3. **Reporter:** Monitors the evolving state of accumulated reasoning and determines the optimal moment to conclude the process, outputting the definitive solution once sufficient validated information has been gathered.

Figure 2 provides an overview of the CR process. As the figure illustrates, CR iteratively refines a solution by progressively integrating validated propositions into the reasoning DAG. While all roles can be instantiated by the same underlying LLM distinguished by role-specific prompts (see Appendix F for detailed examples of prompts and role interactions), the Verifier can also be an external tool. This division of labor aims to overcome limitations of monolithic LLM outputs, where generation and verification are conflated. The specific contribution of each role is crucial: the Proposer explores, the Verifier ensures reliability, and the Reporter synthesizes, all operating on the shared, growing DAG of verified knowledge.

The constructive approach of CR, by building a solution from verified steps, dynamically adjusts the reasoning trajectory. This iterative accumulation and validation of knowledge within the DAG mirrors the nuanced nature of human problem-solving and enhances the robustness of LLMs in complex tasks.

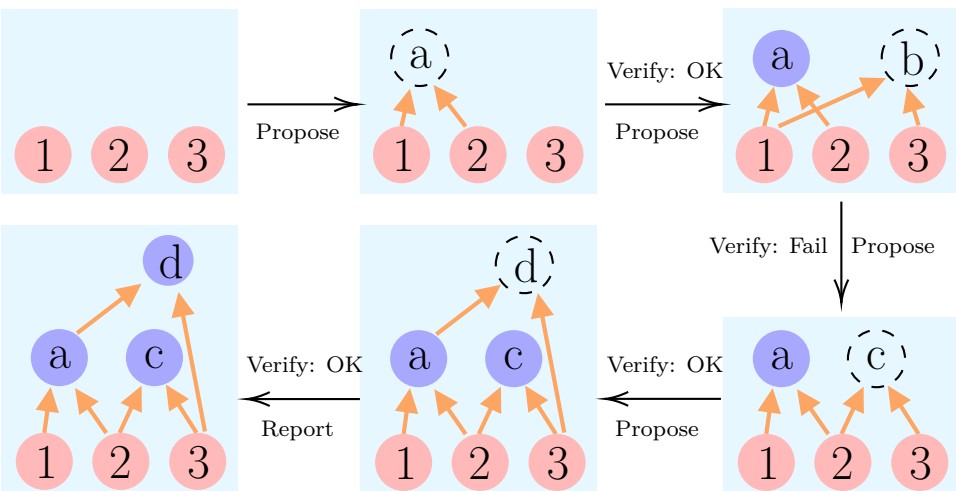

Figure 2: Overview of the Cumulative Reasoning (CR) process applied to a problem with three premises. The diagram illustrates a potential path; in general, new propositions (nodes in the DAG) can be derived by conditioning on any combination of previously validated propositions. The Reporter synthesizes the final answer from the constructed DAG.

## 3.1 Comparison with CoT and ToT

While CR shares the goal of improving multi-step reasoning with Chain-of-Thought (CoT) (Wei et al., 2022) and Tree-of-Thought (ToT) (Yao et al., 2023; Long, 2023) methodologies, its mechanism is distinct. Figure 3 offers a conceptual comparison.

CoT generates a linear sequence of reasoning steps. This approach can be prone to error propagation, as an incorrect intermediate step can derail the entire subsequent chain. It also lacks an explicit mechanism for exploring alternative paths or systematically verifying intermediate conclusions.

ToT extends CoT by exploring multiple reasoning paths in a tree structure, allowing for backtracking and heuristics to guide the search (e.g., breadth-first or depth-first search up to certain limits). While ToT introduces verification or voting at different steps, it typically explores distinct branches which might be pruned, and may not explicitly accumulate all verified knowledge from diverse branches into a single, reusable structure for subsequent reasoning.

CR, in contrast, dynamically constructs a Directed Acyclic Graph (DAG) of all historically validated reasoning steps. This DAG structure allows CR to (i) leverage a more comprehensive and interconnected context of verified information, as new propositions can be derived from any combination of existing validated nodes, not just a linear predecessor or a limited set of ancestors in a tree; (ii) systematically integrate verified knowledge, reducing redundancy and preventing re-exploration of invalid paths due to the Verifier's role. This cumulative and validated knowledge base addresses common issues like error propagation in CoT and potentially fragmented knowledge exploration in ToT.

The explicit Proposer-Verifier-Reporter roles in CR further contribute to a more robust process. This structured decomposition allows for specialized handling of generation, validation, and synthesis, which we hypothesize is more effective than a single model attempting all simultaneously. While conditioning on an increasing number of validated nodes in the DAG can increase contextual complexity, it also provides a richer foundation for subsequent reasoning steps, a trade-off that CR manages through its iterative process.

Other related frameworks like Graph-of-Thought (GoT) (Besta et al., 2024) also explore graph-based reasoning structures. CR offers a specific instantiation where its defined roles iteratively build a Directed Acyclic Graph (DAG). This DAG is crucial: its inherently acyclic structure, combined with CR's focus on cumulative validation at each step, prevents logical loops and ensures consistent forward progression in the deductive process, thereby avoiding reasoning dead ends. We delve deeper into comparisons with other advanced reasoning frameworks in Section 5.

**Detailed Comparison of CoT, ToT, and CR.**  To elucidate the advantages of CR over alternative methods, consider a simplified two-stage reasoning process (which can naturally be extended to multiple stages). For clarity, we assume that whenever a verifier is employed, its accuracy is near-perfect—a condition that can be achieved using symbolic verifier environments (e.g., a Python code interpreter or Lean). We further assume that a unique correct reasoning path exists for the problem. Under these conditions, we define the *arrival probability* as follows. It's important to note that these assumptions, are simplifications for the purpose of this theoretical illustration and may not hold in all practical scenarios.

**Definition 3.1** (Arrival Probability)**.** For a given algorithm, the *arrival probability* is defined as the probability of reaching the correct conclusion from the initial state. Let $P_{\text{CoT}}$ denote the arrival probability for CoT, and $P_{\text{CoT-SC}}$ that for multiple independent CoT trials (self-consistency). Similarly, denote the arrival probability of ToT as

$$P_{\text{ToT}} = p_{1_{\text{ToT}}} \, p_{2_{\text{ToT}}},$$

and that of CR as

$$P_{\text{CR}} = p_{1_{\text{CR}}} \, p_{2_{\text{CR}}},$$

where $p_1$ represents the probability of obtaining the first reasoning step correctly and $p_2$ represents the probability of obtaining the second step correctly, conditioned on the first step being correct.

Since both ToT and CR incorporate verifiers that immediately discard erroneous paths (see Figure 3), it follows that

$$P_{\text{CoT}} \leq p_{1_{\text{ToT}}} \, p_{2_{\text{ToT}}},$$

because CoT, without intermediate verification, is more likely to proceed along an invalid branch.

Note that denoting the arrival probabilities for CR simply as $p_{1_{\text{CR}}}$ or $p_{2_{\text{CR}}}$ is imprecise since CR maintains a history of visited (and validated) states within its DAG. Instead, we write $p_{1_{\text{CR}}|(\mathcal{H})}$ and $p_{2_{\text{CR}}|(\mathcal{H}')}$ to indicate

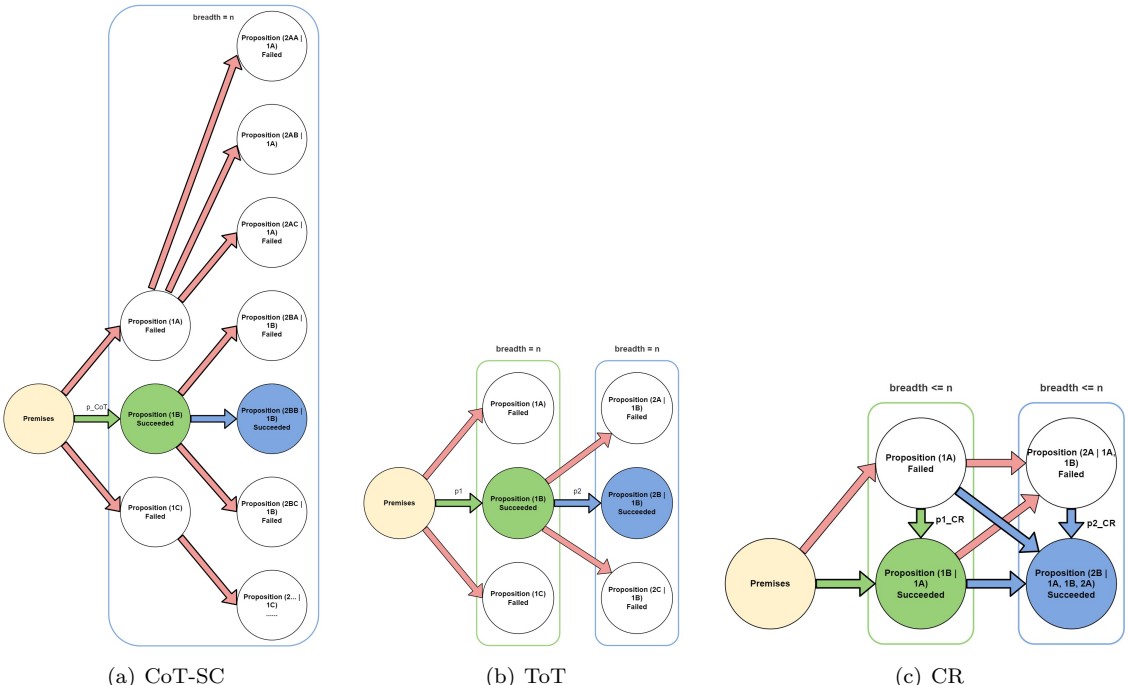

Figure 3: Conceptual comparison between CoT-SC (Self-Consistency on multiple CoT chains), ToT (Tree of Thoughts exploring multiple paths), and CR (Cumulative Reasoning building a DAG of verified steps). CoT-SC generates multiple independent chains and selects the best. ToT explores a tree, potentially pruning branches. CR iteratively builds a DAG, accumulating all verified intermediate steps (green nodes) and using them as context for subsequent reasoning, while invalid steps (red nodes) are discarded by the Verifier.

the probability conditioned on the accumulated history of validated states $\mathcal{H}$ (premises for the first step, premises and validated stage-1 nodes for the second). We make the following assumption, motivated by the intuition that a richer set of verified, relevant information should improve the likelihood of deriving the next correct step, an idea supported by findings in related self-correction and refinement literature (Madaan et al., 2023; Shinn et al., 2023):

**Assumption 3.2.** Given the near-perfect verifier, it holds that for generating a correct step:

$$p_{1_{\mathrm{ToT}}} \leq p_{1_{\mathrm{CR}}|(\cdot)}, \quad p_{2_{\mathrm{ToT}}} \leq p_{2_{\mathrm{CR}}|(\cdot)},$$

and the conditioned probabilities monotonically increase as additional nodes are incorporated:

$$p_{1_{\mathrm{ToT}}} \leq p_{1_{\mathrm{CR}}|(\mathrm{premises})} \leq p_{2_{\mathrm{CR}}|(\mathrm{premises,stage\text{-}1\ node}_1)} \leq p_{2_{\mathrm{CR}}|(\mathrm{premises,stage\text{-}1\ node}_1,\ldots,\mathrm{node}_n)},$$

$$p_{2_{\mathrm{ToT}}} \leq p_{2_{\mathrm{CR}}|(\mathrm{premises,stage\text{-}1\ nodes})} \leq \cdots \leq p_{2_{\mathrm{CR}}|(\mathrm{premises,stage\text{-}1\ nodes,stage\text{-}2\ nodes})}.$$

This assumption posits that CR, by conditioning on a potentially larger set of verified intermediate results from its DAG, is at least as likely, and potentially more likely, to generate the next correct step compared to ToT (which might condition on a more limited history from a single path in its tree). The monotonicity suggests that adding more correct, verified knowledge does not harm the probability of the next step. While LLM behavior can be unpredictable and refinements are not always monotonically beneficial due to issues like hallucination, the presence of a strong verifier in this idealized model helps mitigate such effects for the purpose of this analysis.

The following lemma will be useful for subsequent comparisons.

**Lemma 3.3.** For any positive integer $n$ and any probabilities $p_1, p_2 \in [0, 1]$, the following inequality holds:

$$1 - \left(1 - p_1 \cdot p_2\right)^n \leq \left[1 - \left(1 - p_1\right)^n\right] \cdot \left[1 - \left(1 - p_2\right)^n\right]. \tag{3.1}$$

Please refer to Appendix A for the proof.

**Theorem 3.4** ($P_{\text{CoT-SC}} \leq P_{\text{ToT}} \leq P_{\text{CR}}$)**.** Assume that CoT-SC is executed with $n$ independent trials, while both ToT and CR explore with a maximum breadth of $n$. Under Assumption 3.2, the following inequality holds:

$$P_{\text{CoT-SC}} \leq P_{\text{ToT}} \leq P_{\text{CR}}.P_{\text{CoT-SC}} \leq P_{\text{ToT}} \leq P_{\text{CR}}.$$

To further illustrate the advantages of CR, we conducted a conceptual experiment on the Game of 24. In this experiment, the solution process is divided into two stages. The first stage involves randomly selecting two numbers from the four given inputs to produce a new number, and the second stage employs the remaining three numbers to form an expression that evaluates to 24. We denote the success rates of the first and second stages as $p_1$ and $p_2$, respectively, and let $p$ represent the success rate when solving the Game of 24 directly with the four numbers. The results, summarized in Table 1, indicate that decomposing the problem into sequential steps with intermediate verification significantly improves the overall accuracy.

Table 1: Conceptual experiment results on the Game of 24. The puzzles selected have unique solution paths to facilitate evaluation. Each case was repeated 1000 times.

| Puzzle | $p$ (%) | $p_1$ (%) | $p_2$ (%) | $p_1 p_2$ (%) |
|---|---|---|---|---|
| 2, 7, 12, 13 | 3.0 | 62.3 | 8.0 | 5.0 (+2.0) |
| 6, 11, 12, 13 | 0.0 | 64.8 | 8.0 | 5.2 (+5.2) |
| 8, 8, 10, 12 | 1.8 | 6.9 | 63.9 | 4.4 (+2.6) |

CR's strategy of decomposing problems, meticulously verifying intermediate reasoning steps, and dynamically accumulating validated propositions within a DAG structure offers several advantages. It provides a more robust framework than linear CoT by incorporating verification. Compared to tree-based methods like ToT, CR's cumulative use of all validated knowledge offers a richer contextual basis for subsequent reasoning. These structural and procedural distinctions, grounded in its Proposer-Verifier-Reporter architecture, contribute to CR's strong performance, as suggested by both this illustrative theoretical comparison and the empirical results presented in Section 4. The key insight is that the structured accumulation and reuse of verified knowledge within a flexible DAG enhances complex reasoning.

# 4 Experiments

Our experiments are conducted using the Microsoft Guidance library (Lundberg et al., 2023), which seamlessly integrates generation, prompting, and logical control within language model frameworks. We evaluate our method using the following LLMs: GPT-3.5-turbo, GPT-4, LLaMA-13B, and LLaMA-65B. In our implementation of Cumulative Reasoning (CR), the roles of Proposer, Verifier(s), and Reporter are instantiated using the same underlying LLM but distinguished by role-specific few-shot prompts. This design both broadens the applicability of our approach and simplifies its deployment. Throughout the experiments, we denote by $n$ the number of intermediate propositions generated and by $k$ the number of majority voting iterations. For decoding, we set the temperature to $t = 0.1$ by default and $t = 0.7$ for majority voting. Note that GPT-3.5-turbo and GPT-4 are accessed via OpenAI's chat-format APIs.

## 4.1 FOLIO Wiki

The FOLIO dataset (Han et al., 2022) is a collection of first-order logical inference problems expressed in natural language. Each problem is labeled as "True", "False", or "Unknown." Figure 4 in Appendix D presents an example problem along with solutions generated by both a Chain-of-Thought (CoT) approach and our CR method.

We observe that while the CoT reasoning process may generate useful intermediate steps, it often loses trajectory and fails to reach the correct conclusion. By contrast, CR initially produces two valuable propositions and subsequently leverages them to solve the problem accurately.

The FOLIO dataset is a composite of 1435 examples, of which 52. 5% of these instances have been crafted based on knowledge from randomly selected Wikipedia pages. This approach guarantees the infusion of abundant linguistic variations and a rich vocabulary within the corpus. The residual 47.5% of the examples have been constructed in a hybrid style, based on various complex logical templates. Acknowledging that contemporary LLMs are pre-trained on a considerable volume of human-written corpus, we direct our experiments towards those examples derived from Wikipedia, hereby referred to as FOLIO-wiki. Once a handful of examples are moved aside for few-shot prompts and those examples without source labels for validations are excluded, we are left with a testable collection of 534 examples.

Table 2 reports the performance of different methods evaluated on the FOLIO-wiki dataset. The results demonstrate that CR consistently outperforms Direct prompting, CoT, and CoT with Self-Consistency (CoT-SC), with improvements of up to 8.62%. Notably, when paired with GPT-4, CR achieves an accuracy of 87.45%, compared to 85.02% for GPT-4 with CoT-SC.

Table 2: Results for various reasoning approaches on FOLIO-wiki.

| Model | Method | Acc. ↑ (%) |
|---|---|---|
| - | [Random] | 33.33 |
| LLaMA-13B | Direct
CoT
CoT-SC ($k = 16$)
**CR** ($n = 2$) | 44.75
49.06 (+4.31)
52.43 (+7.68)
**53.37** (+8.62) |
| LLaMA-65B | Direct
CoT
CoT-SC ($k = 16$)
**CR** ($n = 2$) | 67.42
67.42 (+0.00)
70.79 (+3.37)
**72.10** (+4.68) |
| GPT-3.5-turbo | Direct
CoT
CoT-SC ($k = 16$)
**CR** ($n = 2$) | 62.92
64.61 (+1.69)
63.33 (+0.41)
**73.03** (+10.11) |
| GPT-4 | Direct
CoT
CoT-SC ($k = 16$)
**CR** ($n = 2$) | 80.52
84.46 (+3.94)
85.02 (+4.50)
**87.45** (+6.93) |

Table 3: Results for various reasoning approaches on FOLIO-wiki-curated.

| Model | Method | Acc. ↑ (%) |
|---|---|---|
| - | [Random] | 33.33 |
| LLaMA-13B | Direct
CoT
CoT-SC ($k = 16$)
**CR** ($n = 2$) | 49.13
52.17 (+3.04)
53.70 (+4.57)
**55.87** (+6.74) |
| LLaMA-65B | Direct
CoT
CoT-SC ($k = 16$)
**CR** ($n = 2$) | 74.78
74.13 ( -0.65)
79.13 (+4.35)
**79.57** (+4.79) |
| GPT-3.5-turbo | Direct
CoT
CoT-SC ($k = 16$)
**CR** ($n = 2$) | 69.57
70.65 (+1.08)
69.32 ( -0.25)
**78.70** (+9.13) |
| GPT-4 | Direct
CoT
CoT-SC ($k = 16$)
**CR** ($n = 2$) | 89.57
95.00 (+5.43)
96.09 (+6.52)
**98.04** (+8.47) |

Table 4: Experimental results on the FOLIO-wiki and FOLIO-wiki-curated datasets.

## 4.2 FOLIO Wiki Curated

A detailed review of the FOLIO-wiki dataset revealed several problematic instances, including: 1) Missing or contradictory common knowledge; 2) Overly ambiguous problems that do not yield unequivocal answers; 3) Inherent inconsistencies within the premises; 4) Vague statements or typographical errors; 5) Incorrect answer annotations.

After removing 74 such problematic instances, the curated set comprises 460 examples (see Appendix E.2 for detailed examples). As shown in Table 3, when applied to this refined dataset, GPT-4 paired with CR achieves an accuracy of 98.04% (error rate: 1.96%), nearly doubling the effectiveness of GPT-4 with CoT-SC.

## 4.3 AutoTNLI

**Experimental Setting.** The AutoTNLI dataset (Kumar et al., 2022) extends the INFOTABS dataset (Gupta et al., 2020) to construct a challenging Tabular Natural Language Inference task. This dataset contains 1,478,662 table-hypothesis pairs labeled as either "Entail" or "Neutral." In our adaptation, we treat the tabular data as premises in a manner analogous to the FOLIO dataset. Due to the dataset's large scale,

we limit our evaluation to the first 1,000 table-hypothesis pairs and compare the performance of LLaMA-13B and LLaMA-65B using Direct, CoT, CoT-SC, and our CR method.

**Evaluation Results.** Table 6 shows that CR significantly outperforms the alternative prompting strategies. In particular, LLaMA-65B with CR achieves a 12.8% accuracy improvement over CoT-SC, demonstrating CR's superior ability to capture structural and linguistic nuances in logical inference.

**More Experiments and Ablation Studies.** Regarding the computational complexity of different methods, Table 13 and Table 14 (please refer to Appendix B) demonstrate the superiority of CR over CoT, CoT-SC, and ToT on several logical inference tasks, including the LogiQA (Liu et al., 2020) and ProofWriter (Tafjord et al., 2020) datasets. In addition, Table 5 presents ablation studies on the FOLIO wiki dataset using the GPT-3.5-turbo model, quantifying the impact of individual components—such as the verifier and the premises random choice mechanism—on CR's performance.

Table 5: Ablation studies on FOLIO wiki dataset using GPT-3.5-turbo model.

| Model | Method | Acc. ↑ (%) |
|---|---|---|
| - | [Random] | 33.33 |
| GPT-3.5-turbo | Direct | 62.92 |
| | CoT | 64.61 (+1.69) |
| | CoT-SC (k = 16) | 63.33 (+0.41) |
| | **CR (ours**, $n = 2$) | **73.03 (+10.11**) |
| | **CR (ours**, $n = 2$, w/o Verifier) | 64.23 (+1.31) |
| | **CR (ours**, $n = 2$, w/o premises random choice) | 68.73 (+5.81) |
| | **CR (ours**, $n = 2$, w/o Verifier, w/o premises random choice) | 67.23 (+4.31) |

Table 6: Results on the AutoTNLI dataset.

| Model | Method | Acc. ↑ (%) |
|---|---|---|
| - | [Random] | 50.00 |
| LLaMA-13B | Direct | 52.6 |
| | CoT | 54.1 (+1.5) |
| | CoT-SC ($k = 16$) | 52.1 (-0.5) |
| | **CR** ($n = 4$) | **57.0** (+5.4) |
| LLaMA-65B | Direct | 59.7 |
| | CoT | 63.2 (+3.5) |
| | CoT-SC ($k = 16$) | 61.7 (+2.0) |
| | **CR** ($n = 4$) | **72.5** (+12.8) |

Table 7: Results on the Game of 24 using GPT-4.

| Method | Acc. ↑ (%) | # Visited States ↓ |
|---|---|---|
| Direct | 7.3 | 1 |
| CoT | 4.0 | 1 |
| CoT-SC (k=100) | 9.0 | 100 |
| Direct (best of 100) | 33 | 100 |
| CoT (best of 100) | 49 | 100 |
| ToT (b=5) | 74 | 61.72 |
| **CR** (b=1) | 84 (+10) | **11.68** (-50.04) |
| **CR** (b=2) | 94 (+20) | 13.70 (-48.02) |
| **CR** (b=3) | 97 (+23) | 14.25 (-47.47) |
| **CR** (b=4) | 97 (+23) | 14.77 (-46.95) |
| **CR** (b=5) | **98** (+24) | 14.86 (-46.86) |

Table 8: Experimental results on the AutoTNLI and Game of 24 datasets.

## 4.4 Game of 24

The Game of 24 is a numerical puzzle in which players must combine four given integers using basic arithmetic operations (addition, subtraction, multiplication, and division) to yield 24. A puzzle is considered successfully solved if the resulting equation is valid and each input number is used exactly once.

**Experimental Setup.** We evaluate a set of 100 puzzles curated by ToT (Yao et al., 2023). Our primary metrics are accuracy and the average number of visited states during the search. In our CR algorithm, a set of reachable states, $S$, is maintained. The process begins with the initial state $s$ (the four input numbers without operations), and at each iteration a state $u \in S$ is selected. The Proposer chooses two numbers from $u$ and applies a basic operation to generate a new state $v$. After the Verifier confirms the validity of the operation, $v$ is added to $S$. When a state representing a valid solution (i.e., an equation that evaluates to 24) is reached, the Reporter traces back the derivation and outputs the solution. The process terminates

either when a solution is reported or when the iteration count exceeds a predefined limit ($L = 50$). We run multiple parallel branches (with breadth $b$ ranging from 1 to 5) to account for variability in the search.

**Results.** As summarized in Table 7, CR substantially outperforms ToT by achieving up to 98% accuracy (a 24% improvement over ToT's 74%) while exploring significantly fewer states.

**Comparison with ToT.** In the specific context of the Game of 24, the methodologies of Cumulative Reasoning (CR) and Tree of Thoughts (ToT) share similarities yet diverge significantly in their approach to state generation and exploration. A fundamental difference lies in how each iteration processes: CR is designed to introduce a single new state at each step, focusing on a step-by-step progression towards the solution. Conversely, ToT is characterized by its generation of multiple candidate states during each iteration, employing a filtration mechanism to narrow down the feasible states. This operational distinction suggests that ToT engages in a broader exploration of potential, including invalid, states, compared to the more streamlined approach of CR.

Furthermore, ToT relies on a pre-defined search structure, utilizing a constant width and depth within its search tree. This rigid framework contrasts with CR's more dynamic strategy, where the language model (LLM) itself influences the depth of the search, adapting the exploration breadth as needed across different stages of the problem-solving process. Such flexibility in CR not only optimizes the search path but also tailors the exploration to the complexity and requirements of each specific problem, potentially enhancing efficiency and efficacy in reaching the correct solution.

Table 9: Comparative performance on the MATH dataset using GPT-4 without code environment. We adopted a default temperature setting of $t = 0.0$, consistent with prior research settings (greedy decoding). PHP denotes the application of the progressive-hint prompting. "Iters" represents the average number of LLM interactions, and **Overall** reflects the overall results across MATH subtopics (* denotes using 500 test examples subset following Lightman et al. (2023)).

| | w/ PHP | MATH Dataset | | | | | | |
|---|---|---|---|---|---|---|---|---|
| | | Precalc | Geometry | NumT | Prob | PreAlg | Algebra | **Overall** |
| CoT | ✗ | - | - | - | - | - | - | 42.50 |
| Complex CoT | ✗ | 26.7 | 36.5 | 49.6 | 53.1 | 71.6 | 70.8 | 50.36 |
| 8 shot | ✓ | 29.8 | 41.9 | 55.7 | 56.3 | 73.8 | 74.3 | 53.90 |
| | (Iters) | 3.2435 | 3.2233 | 3.1740 | 2.8122 | 2.3226 | 2.4726 | 2.8494 |
| Complex CoT* | ✗ | 33.9 | 34.1 | 46.8 | 47.4 | 62.1 | 70.7 | 48.80 |
| (repro., 8-shot) | ✓ | 30.4 | 43.9 | 53.2 | 50.0 | 68.5 | 84.1 | 53.80 |
| | (Iters) | 2.4643 | 2.7805 | 2.7581 | 2.4474 | 2.3780 | 2.5484 | 2.59 |
| **CR w/o code*** | ✗ | 30.4 (-3.5) | 39.0 (**+4.9**) | 54.8 (**+8.0**) | 57.9 (**+10.5**) | 71.8 (**+9.7**) | 79.3 (**+8.6**) | **54.20 (+5.40)** |
| (**ours**, 4-shot) | ✓ | **35.7 (+5.3)** | **43.9 (+0.0)** | **59.7 (+6.5)** | **63.2 (+13.2)** | **71.8 (+3.3)** | **86.6 (+2.5)** | **58.00 (+4.20)** |
| | (Iters) | 2.4821 | 2.5122 | 2.2903 | 2.2105 | 2.2195 | 2.3548 | **2.40 (-0.19)** |

## 4.5 Solving MATH Problems

The MATH dataset (Hendrycks et al., 2021) provides a comprehensive benchmark for mathematical reasoning across diverse subdomains such as Algebra and Geometry. We compare Complex CoT and our CR method, both with and without Progressive-Hint Prompting (PHP) (Zheng et al., 2023). Our reproduction follows the evaluation protocol of Lightman et al. (2023), using an 8-shot prompting strategy on a 500-example subset that spans all difficulty levels (Levels 1–5).

**Results without Code Environment.** Table 9 shows that CR outperforms Complex CoT by 5.4% in overall accuracy when using a 4-shot strategy. In particular, CR yields substantial gains in Number Theory, Probability, PreAlgebra, and Algebra. Table 10 further demonstrates that at Level 5—the most challenging subset—CR achieves a 9.7% improvement, corresponding to a 43% relative gain over Complex CoT without PHP. The "Iters" column in Table 9 indicates the average number of LLM interactions, providing insight into the computational effort. CR demonstrates competitive iteration counts while achieving higher accuracy.

**With a Code Environment.** In addition to the experiments above, we extend CR by incorporating a Python code environment to emulate a semi-symbolic reasoning system. In this configuration, no external

Table 10: Comparative performance on the MATH dataset using GPT-4 without code environment for different difficulty levels. (* denotes evaluation on a 500-example subset.)

| | w/ PHP | MATH Dataset (* denotes using 500 test examples subset) | | | | | |
| --- | --- | --- | --- | --- | --- | --- | --- |
| | | Level 5 | Level 4 | Level 3 | Level 2 | Level 1 | **Overall** |
| CoT (OpenAI, 2023) | ✗ | - | - | - | - | - | 42.50 |
| Complex CoT* | ✗ | 22.4 | 38.3 | 62.9 | 72.2 | 79.1 | 48.80 |
| (repro., 8-shot) | ✓ | 23.9 | 43.8 | 63.8 | 86.7 | 83.7 | 53.80 |
| **CR w/o code*** | ✗ | **32.1** (+9.7) | 43.0 (+4.7) | 62.9 (+0.0) | 78.9 (+6.7) | 83.7 (+4.6) | **54.20** (+5.40) |
| (**ours**, 4-shot) | ✓ | 27.3 (+3.4) | **50.0** (+6.2) | **70.9** (+7.1) | **86.7** (+0.0) | **90.7** (+7.0) | **58.00** (+4.20) |

aids (such as memory modules, web Browse, or retrieval systems) are employed. Instead, the Python interpreter serves as a highly reliable and efficient **Verifier**, executing and validating arithmetic or symbolic expressions generated by the LLM Proposer. This setup allows the Proposer to generate hypotheses and mathematical expressions, which are then rigorously verified through code execution before being added to the CR's knowledge DAG. This highlights CR's flexibility in integrating different types of verifiers.

Tables 11 and 12 compare our approach with state-of-the-art methods such as PAL (Gao et al., 2023) and ToRA (Gou et al., 2023). CR with code achieves an overall accuracy of 72.2% on the MATH dataset, reflecting a 38.9% relative improvement over PAL and an 18.8% improvement over ToRA. Furthermore, on the hardest Level 5 problems, CR with code exhibits a 66.8% relative improvement over PAL and a 12.8% improvement over ToRA.

Table 11: Comparative performance on the MATH dataset using GPT-4 with a Python code environment. Results are based on greedy decoding ($t = 0.0$). **Overall** reflects the aggregate accuracy across MATH subtopics.

| | Precalc | Geometry | NumT | Prob | PreAlg | Algebra | **Overall** |
| --- | --- | --- | --- | --- | --- | --- | --- |
| PAL | 29.3 | 38.0 | 58.7 | 61.0 | 73.9 | 59.1 | 51.8 |
| PAL* (repro., 4-shot) | 23.2 | 31.7 | 66.1 | 57.9 | 73.2 | 65.3 | 52.0 |
| ToRA | 37.2 | 44.1 | 68.9 | 67.3 | 82.2 | 75.8 | 61.6 |
| ToRA* (repro., 4-shot) | 44.6 | 48.8 | 49.5 | 66.1 | 67.1 | 71.8 | 60.8 |
| **CR w/ code*** (ours, 2-shot) | **51.8** (+7.2) | **53.7** (+4.9) | **88.7** (+22.6) | **71.1** (+5.0) | **86.6** (+13.4) | **86.3** (+14.5) | **72.2** (+11.4) |

Table 12: Comparative performance on the MATH dataset using GPT-4 with a Python code environment across different difficulty levels.

| | Difficulty Levels | | | | | |
| --- | --- | --- | --- | --- | --- | --- |
| | Level 5 | Level 4 | Level 3 | Level 2 | Level 1 | **Overall** |
| PAL | - | - | - | - | - | 51.8 |
| PAL* (repro., 4-shot) | 31.3 | 45.3 | 60.0 | 65.6 | 88.4 | 52.0 |
| ToRA | - | - | - | - | - | 61.6 |
| ToRA* (repro., 4-shot) | 46.3 | 53.9 | 69.5 | 75.6 | 74.4 | 60.8 |
| **CR w/ code*** (ours, 2-shot) | **52.2** (+5.9) | **66.4** (+12.5) | **81.9** (+12.4) | **90.0** (+14.4) | **90.7** (+2.3) | **72.2** (+11.4) |

## 5 Related Work

**Reasoning with Large Language Models (LLMs).** The quest to imbue LLMs with robust reasoning capabilities has spurred extensive research. Early approaches focused on generating intermediate steps (Zaidan et al., 2007; Yao et al., 2021; Hase & Bansal, 2021; Yang et al., 2022; Wu et al., 2022; Zhou et al., 2022). Morishita et al. (2023) enhance language models' reasoning by utilizing a synthetic corpus based on formal logic theory. Uesato et al. (2022); Lightman et al. (2023) compare process-based and outcome-based

approaches in solving mathematical reasoning tasks. Further, a considerable breadth of research focuses on augmenting reasoning through symbolic systems, such as code environments, knowledge graphs, and formal theorem provers, showcasing the utility of hybrid approaches in complex reasoning tasks (Mihaylov & Frank, 2018; Bauer et al., 2018; Kundu et al., 2018; Wang et al., 2019; Lin et al., 2019; Ding et al., 2019; Feng et al., 2020; Nye et al., 2021; Wang et al., 2022a; Chen et al., 2022; Lyu et al., 2023; Chen et al., 2022; Gao et al., 2023; Gou et al., 2023; Li et al., 2023a; Jiang et al., 2022; Yang et al., 2023).

**Chain-of-Thought (CoT) Prompting.** Initiated by Wei et al. (2022), the CoT reasoning paradigm underscores the value of multi-step logical pathways in deriving conclusive answers. Building on this, Wang et al. (2022b) introduce self-consistency as an advanced decoding strategy, aiming to refine the basic greedy decoding used in CoT. Zhou et al. (2022) (Least-to-Most) and Khot et al. (2022) (Decomposed Prompting) further dissect complex tasks into manageable sub-tasks. Creswell & Shanahan (2022) enhance reasoning quality via beam search. Fu et al. (2022) (Complex CoT) advocate for more complex few-shot prompts. Creswell & Shanahan (2022) explore the enhancement of reasoning quality through a beam search across reasoning traces, while Fu et al. (2022) argue for increasing the complexity within few-shot prompts to improve performance. Recent developments include Li et al. (2023b)'s DIVERSE, which investigates various reasoning paths for the same question and employs a verifier for accuracy through weighted voting. Du et al. (2023) present a multi-agent debate approach with multiple LLMs. Yao et al. (2023)'s Tree-of-Thought (ToT) framework introduces deliberation in decision-making by considering multiple reasoning paths. Zheng et al. (2023) propose an iterative approach, using previous responses as contextual clues in subsequent iterations. Feng et al. (2023) highlight the theoretical and practical implications of CoT for solving complex real-world tasks, including dynamic programming. Recently, there have also been many works on the self-criticizing process (Tyen et al., 2023; Li et al., 2024; Zhang et al., 2024b; Lin et al., 2024; Wang et al., 2024), showing that language models can have self-correction capabilities with theoretical guarantees.

**Advanced Reasoning Frameworks.** Yao et al. (2023); Long (2023)'s Tree-of-Thought (ToT) framework allows LLMs to explore multiple reasoning paths and self-evaluate choices. As discussed in Section 3.1, CR differs from ToT by its cumulative DAG construction and distinct Proposer-Verifier-Reporter roles, fostering a more integrated use of all validated knowledge. Graph-of-Thoughts (GoT) frameworks (Besta et al., 2024) also leverage graph structures for reasoning, representing thoughts as nodes and dependencies as edges. CR offers a specific operationalization of graph-based reasoning with its iterative, role-based DAG construction and explicit verification at each step. Forest of Thought (FoT) methodologies (Bi et al., 2024) may explore multiple diverse reasoning trees or high-level plans concurrently. CR, while capable of exploring branches (e.g., parameter $b$ in Game of 24), primarily focuses on the meticulous, cumulative construction of a single, coherent reasoning DAG.

**Self-Critique and Iterative Refinement.** CR's Verifier role aligns with and extends self-critique concepts (Madaan et al., 2023; Shinn et al., 2023; Tyen et al., 2023; Hosseini et al., 2024; Li et al., 2024; Lin et al., 2024; Wang et al., 2024; Zhang et al., 2024b). While many self-critique methods refine a single reasoning trace or select among alternatives (e.g., Reflexion (Shinn et al., 2023), Self-Refine (Madaan et al., 2023)), CR uses verification to build a persistent, growing DAG of validated knowledge that informs all subsequent reasoning steps. This iterative accumulation and reuse of verified steps is a key distinction. Zheng et al. (2023) (Progressive-Hint Prompting) also uses an iterative approach with previous responses as context, which CR incorporates in its MATH experiments.

Recent advancements focus on more nuanced verifiers and verification frameworks. For example, Li et al. (2023b)'s DIVERSE employs a verifier for accuracy through weighted voting over various reasoning paths; in contrast, CR's verification is more deeply integrated into the step-by-step construction of the reasoning DAG. Hosseini et al. (2024) propose V*, which improves LLM's reasoning by training a verifier on both correct and incorrect solutions generated during self-improvement, which then helps select the best answer from multiple candidates at inference time. More recent approaches, such as Li et al. (2025) (PANEL), investigate detailed, stepwise natural language self-critique providing linguistic feedback, while Ma et al. (2025) (General-Reasoner) introduce a generative model-based verifier. While CR's current Verifier often

makes binary (valid/invalid) decisions or relies on external tools like code interpreters, future work could integrate such richer feedback mechanisms into the Verifier role.

## 6 Conclusion

In this work, we introduce Cumulative Reasoning (CR), an approach leveraging LLMs in a structured, iterative process that mirrors human cognitive strategies. By orchestrating the roles of proposer, verifier(s), and reporter, CR not only decomposes complex problems into manageable tasks but also effectively recomposes the validated steps into comprehensive solutions. This methodology has demonstrated superior performance across various domains, including logical inference, the Game of 24, and MATH problems, showcasing the versatility and potential of CR in advancing the capabilities of LLMs in complex problem-solving scenarios.

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

# Appendix

# A Proofs of Theorems

**Proof of Lemma 3.3.**

*Proof.*

$$1 - (1 - p_1 \cdot p_2)^n \leq (1 - (1 - p_1)^n) \cdot (1 - (1 - p_2)^n)$$
$$\Leftrightarrow 1 - (1 - p_1 \cdot p_2)^n \leq 1 - (1 - p_1)^n - (1 - p_2)^n + (1 - p_1)^n \cdot (1 - p_2)^n$$
$$\Leftrightarrow (1 - p_1)^n + (1 - p_2)^n \leq (1 - p_1 \cdot p_2)^n + (1 - p_1)^n \cdot (1 - p_2)^n$$
$$\Leftrightarrow (1 - p_1)^n + (1 - p_2)^n \leq (1 - p_1 \cdot p_2)^n + (1 - p_1 - p_2 + p_1 \cdot p_2)^n$$

Notice that

$$(1 - p_1 \cdot p_2) + (1 - p_1 - p_2 + p_1 \cdot p_2) \equiv (1 - p_2) + (1 - p_2) \equiv 2 - p_1 - p_2,$$

WLOG, let $p_1 \geq p_2$ , then

$$(1 - p_1 - p_2 + p_1 \cdot p_2) \leq (1 - p_1) \leq (1 - p_2) \leq (1 - p_1 \cdot p_2).$$

From the monotonicity of function $x^n + (2 - p_1 - p_2 - x)^n$ in the interval $(-\infty, \frac{2-p_1-p_2}{2}]$ and the interval $[\frac{2-p_1-p_2}{2}, +\infty)$ respectively, and the symmetry of $\{(1 - p_1 - p_2 + p_1 \cdot p_2), (1 - p_1 \cdot p_2)\}$ and the symmetry of $\{(1 - p_1), (1 - p_2)\}$ correspond to $y = \frac{2-p_1-p_2}{2}$, we conclude the proof. □

**Proof of Theorem 3.4.**

*Proof.*

$$P_{\text{CoT-SC}} \leq 1 - (1 - p_{\text{CoT}})^n \leq 1 - (1 - p_1 \cdot p_2)^n,$$

$$P_{\text{ToT}} = (1 - (1 - p_1)^n) \cdot (1 - (1 - p_2)^n),$$

Combined with Lemma 3.3, now we have

$$P_{\text{CoT-SC}} \leq P_{\text{ToT}}.$$

From Assumption 3.2, we have

$$P_{\text{ToT}} \leq (1 - (1 - p_{1_{\text{CR}|(\text{premises})}})^n) \cdot (1 - (1 - p_{2_{\text{CR}|(\text{premises, stage-1 nodes})}})^n) \leq P_{\text{CR}}.$$

Finally, we conclude that

$$P_{\text{CoT-SC}} \leq P_{\text{ToT}} \leq P_{\text{CR}}.$$

□

# B More on Experiments

## B.1 More Experimental Results

For a fair comparison of different methods on the LogiQA, ProofWriter, FOLIO (validation set), and LD datasets, we report the third-party reproduced results by Sun et al. (2023). For details on the implementation of these experiments, please refer to their work.

Table 13: Comparison results on LogiQA

| Method | Acc. ↑ | # Visited States ↓ |
|---|---|---|
| Direct | 31.69% | 1 |
| CoT | 38.55% | 1 |
| CoT-SC | 40.43% | **16** |
| ToT | 43.02% | 19.87 |
| CR | **45.25**% | 17 |

Table 14: Comparison results on ProofWriter

| Method | Acc. ↑ | # Visited States ↓ |
|---|---|---|
| Standard | 46.83% | 1 |
| CoT | 67.41% | 1 |
| CoT-SC | 69.33% | **16** |
| ToT | 70.33% | 24.57 |
| CR | **71.67**% | 16.76 |

Table 15: Comparison results on FOLIO-val

| Method | Acc. ↑ | # Visited States ↓ |
|---|---|---|
| Standard | 60.29% | 1 |
| CoT | 67.65% | 1 |
| CoT-SC | 68.14% | 16 |
| ToT | **69.12**% | 19.12 |
| CR | **69.11**% | **15.87** |

Table 16: Comparison results on LD

| Method | Acc. ↑ | # Visited States ↓ |
|---|---|---|
| Standard | 71.33% | 1 |
| CoT | 73.33% | 1 |
| CoT-SC | 74.67% | **16** |
| ToT | 76.83% | 21.83 |
| CR | **78.33**% | 16.98 |

Table 17: Comparison of results on different datasets.

## B.2 Computational Considerations

The structured approach of CR, involving distinct Proposer, Verifier, and Reporter roles, and iterative DAG construction, may entail different computational overhead compared to simpler methods like CoT or even ToT, depending on the configuration.

If all roles are LLM-based, CR can involve more LLM calls per problem than a single CoT pass. For instance, each reasoning cycle might involve a Proposer call and a Verifier call. However, this is a deliberate design choice aiming for higher accuracy. The number of iterations is managed by the Reporter or a predefined limit. As shown in our experiments (e.g., "Iters" in Table 9 for MATH problems, and "#Visited States" in Table 7 for Game of 24, and additional data in Appendix B such as Table 13 and 14), CR often achieves superior accuracy. For example, in the Game of 24 (Table 7), CR (b=5) achieved 98% accuracy with only 14.86 visited states, while ToT (b=5) achieved 74% with 61.72 states. This suggests that while individual steps in CR might be more involved due to verification, the overall search can be more efficient by avoiding unproductive paths.

A significant aspect of CR's flexibility is the Verifier (Hosseini et al., 2024). When an LLM is used as a Verifier (as in some of our FOLIO experiments), it adds to the LLM call count. However, as demonstrated in our MATH experiments with a code environment (Section 4.5), the Verifier can be a non-LLM tool (e.g., a Python interpreter). Such symbolic verifiers are typically much faster and cheaper than LLM calls, significantly reducing the overhead of the verification step while increasing its reliability.

CR's approach of conditioning on previously validated steps in the DAG means the context provided to the Proposer can grow. While this provides richer information, it can also increase the token count for LLM calls. Future work could explore optimizing these dependencies, for instance, by selecting only the most relevant prior steps to manage context size without sacrificing much of the benefit of cumulative knowledge.

CR trades potentially increased computational steps (especially if LLM verifiers are used) for significant gains in accuracy and robustness, particularly on complex tasks. Its efficiency in terms of exploring fewer states to reach a correct solution, as seen in several benchmarks, suggests that it offers a favorable balance, making it suitable for scenarios where high performance is critical, potentially being more effective than exhaustive search or very wide ToT explorations for achieving similar results. The architecture uses the same underlying LLM for all roles (distinguished by prompts), avoiding the need to load multiple large

models. We believe CR is particularly well-suited for test-time scaling where achieving the best possible answer justifies the additional computational steps.

## C   More on Logic

**Limitations of First-Order Logic Systems.** It is not surprising that the labels verified by FOL are still not satisfying. There are several limitations inside the FOL systems:

1. Limitations of Expressiveness (Löwenheim, 1967): FOL even lacks the expressive power to capture some properties of the real numbers. For example, properties involving uncountably many real numbers often cannot be expressed in FOL. In addition, properties requiring quantification over sets of real numbers or functions from real numbers to real numbers cannot be naturally represented in FOL.

2. Translation Misalignment: Risk of semantic discrepancies during translation, rendering resolutions ineffective. For instance, translating statements as $\forall \text{Bird}(x) \Rightarrow \text{CanFly}(x)$ and $\forall x(\text{Fly}(x) \Rightarrow \text{Wings}(x))$ may cause a misalignment between "CanFly" and "Fly", leading to flawed conclusions. It often fails to capture the full richness and ambiguity of natural language and lacks basic common knowledge (Gamut, 1990).

3. Undecidability: The general problem of determining the truth of a statement in FOL is undecidable (Turing et al., 1936; Chimakonam, 2012) (deeply connected to the halting problem), constraining its applicability for automated reasoning in complex tasks.

### C.1   Illustrative example on higher-order logic

Here we present a refined example derived from the FraCas dataset to illustrate higher-order logic inference. It is noteworthy that the FraCas dataset (Cooper et al., 1996) is dedicated to the realm of higher-order logic inference. This characterization also applies to a majority of the Natural Language Inference (NLI) datasets (Kumar et al., 2022), which encompass their internal syntax, semantics, and logic. The intricate linguistic components such as quantifiers, plurals, adjectives, comparatives, verbs, attitudes, and so on, can be formalized with Combinatory Categorial Grammar (CCG) along with the formal compositional semantics (Mineshima et al., 2015).

Higher-order logic (HOL) has the following distinctive characteristics as opposed to FOL (Mineshima et al., 2015):

**Quantification over Functions**: Higher-order logic (HOL) allows for lambda expressions, such as $\lambda y.\text{report\_attribute}(y, \text{report})$, whereby functions themselves become the subject of quantification. An illustration of this is found in the expression "a representative who reads this report." Here, quantification spans the predicates representing both the representative and the reading of the report, a phenomenon captured as a higher-order function. Unlike HOL, FOL is incapable of extending quantification to functions or predicates.

**Generalized Quantifiers**: The introduction of generalized quantifiers, such as "most," serves as another demarcation line between HOL and FOL. These quantifiers are capable of accepting predicates as arguments, enabling the representation of relations between sets, a feat that transcends the expressive capacity of FOL.

**Modal Operators**: Employing modal operators like "might" signifies a transition towards HOL. These operators, applicable to propositions, give rise to multifaceted expressions that defy easy reduction to the confines of FOL.

**Attitude Verbs and Veridical Predicates**: The integration of attitude verbs, such as "believe," and veridical predicates like "manage," injects an additional layer of complexity necessitating the use of HOL. These linguistic constructs can engage with propositions as arguments, interacting with the truth values of those propositions in subtle ways that demand reasoning extending beyond the capabilities of FOL.

Previously we have discussed the limitations of FOL systems, what about HOL systems? Crafting HOL programs that are solvable by symbolic systems is a daunting task, even for experts. It is also challenging for LLMs to write these intricate programs effectively. Using formal theorem provers based on higher-order (categorical) logic and (dependent) type theory ups the ante, making it even harder. However, CR solves

these problems pretty well without resorting to and being restricted to symbolic systems, just like the way humans think.

---

**[Modified Example FraCas-317]**

- **Premises**:
    1. Most of the representatives who read the report have a positive attitude towards it.
    2. No two representatives have read it at the same time, and they may have different opinions about it.
    3. No representative took less than half a day to read the report.
    4. There are sixteen representatives.
- **Hypothesis**: It took the representatives more than a week to read the report, and most found it valuable.
- **Label**: [**True**]
- **Higher-Order Logic Premises**:
    1.

$$\text{most}(\lambda x.\text{representative}(x) \wedge \text{reads}(x, \text{report}), \lambda x.\text{has\_positive\_attitude}(x, \text{report}))$$

    2.

$$\neg \exists x, y \, (x \neq y \wedge \text{representative}(x) \wedge \text{representative}(y) \wedge$$
$$\text{read\_at\_same\_time}(x, y, \text{report})) \wedge$$
$$\forall x, y \, (x \neq y \wedge \text{representative}(x) \wedge \text{representative}(y) \rightarrow$$
$$\text{may}(\lambda z.\text{different\_opinions}(z, x, y, \text{report})))$$

    3. $\neg \exists x(\text{representative}(x) \wedge \text{took\_less\_than\_half\_day}(\lambda y.\text{read}(y, \text{report})))$
    4. $\text{count}(\lambda x.\text{representative}(x)) = 16$
- **Higher-Order Logic Hypothesis**:

$$\exists t(\text{total\_reading\_time}(\text{report}) > 7 \wedge \text{time\_unit}(t, \text{day})) \wedge$$
$$\text{most}(\lambda x.\text{representative}(x), \lambda x.\text{found\_valuable}(x, \text{report}))$$

---

**[Cumulative Reasoning by GPT-4]**

- **Generated Propositions**:
    1. "Using P1 and P2, we can deduce Q1: Every representative read the report at a different time, and most representatives found the report valuable."
    2. "Using P3 and Q1, we deduce Q2: The total reading time must be at least 8 days. (since 16 representatives each took at least half a day)."
    3. "Using Q2 and P4, we deduce Q3: The representatives took at least 8 days to read the report."
- **Reasoning**: "Based on the intermediate propositions Q1, Q2, and Q3, the hypothesis that it took the representatives more than a week to read the report, and most found it valuable, is confirmed as True."
- **Prediction**: [**True**] (**Correct**)

---

## D  Appendix for Examples

---

**[Problem Description]**

**Premises:**

(1) It is not true that some giant language models do not have good performance.

(2) All language models with good performance are used by some researchers.

(3) If a language model is used by some researchers, it is popular.

(4) If BERT is a giant language model, then GPT-3 is also a giant language model.

(5) BERT is a giant language model.

**Hypothesis**: GPT-3 is popular.

**Label**: [**True**]

---

**[Chain-of-Thought Reasoning by GPT-4]**

**Reasoning**: From the premises, we have:

1. $\neg(\exists x)(\text{Giant}(x) \wedge \neg\text{GoodPerformance}(x))$
2. $\forall x\,(\text{GoodPerformance}(x) \rightarrow \text{UsedByResearchers}(x))$
3. $\forall x\,(\text{UsedByResearchers}(x) \rightarrow \text{Popular}(x))$
4. $\text{BERT} \rightarrow \text{GPT-3}$
5. $\text{BERT}$

Given that BERT is a giant language model, premise (4) implies that GPT-3 is also a giant language model. However, the premises do not explicitly provide information regarding GPT-3's performance or usage by researchers, and hence no definitive conclusion on its popularity can be drawn.

**Prediction**: [**Unknown**] (**Incorrect**)

---

**[Cumulative Reasoning by GPT-4]**

**Generated Propositions**:

(1) GPT-3 is a giant language model with good performance.

(2) GPT-3 is used by some researchers.

**Reasoning**: Since GPT-3 is a giant language model with good performance and is used by researchers, and given that any language model used by researchers is considered popular, it follows that GPT-3 is popular.

**Prediction**: [**True**] (**Correct**)

---

Figure 4: Example from the FOLIO dataset. The left panel shows a problem along with its premises, while the subsequent panels display reasoning by CoT and CR respectively. CR leverages intermediate propositions to yield the correct prediction.

---

**[Illustrative example for Game of 24]**

- Numbers: [4, 5, 6, 10]
- Arithmetic Operations: $[+, -, \times, /, (, )]$
- **Solution**:
$$(10 - 6) * 5 + 4 = 24$$

---

Figure 5: An example from the Game of 24 dataset (Yao et al., 2023).

[**Problem Description**]

- Example ID: test/intermediate_algebra/1350.json

- Level: 5

- Subject: Intermediate Algebra

- **Problem**: Consider the polynomial

$$f(x) = a_n x^n + a_{n-1} x^{n-1} + \cdots + a_2 x^2 + a_1 x + a_0,$$

  where the polynomial has integer coefficients and its roots are distinct integers.

  Given $a_n = 2$ and $a_0 = 66$, the inquiry is to determine the least possible value of $|a_{n-1}|$.

[**Ground Truth Solution**]

- **Solution**: Since $f(x)$ has integer coefficients, the Integer Root Theorem asserts that any integer roots of $f(x)$ must divide the constant term $66 = 2 \cdot 3 \cdot 11$. Consequently, the potential integer roots of $f(x)$ are

$$\pm 1, \ \pm 2, \ \pm 3, \ \pm 6, \ \pm 11, \ \pm 22, \ \pm 33, \ \pm 66.$$

  Additionally, given that all roots of $f(x)$ are integers, they are necessarily members of the aforementioned list.

  We proceed to utilize Vieta's formulas. The roots of $f(x)$ yield a product of $(-1)^n \cdot \frac{a_0}{a_n}$, which evaluates to either 33 or $-33$. Simultaneously, the sum of these roots is $-\frac{a_{n-1}}{a_n} = -\frac{a_{n-1}}{2}$. To minimize $|a_{n-1}|$, we aim to reduce the absolute value of the root sum, ensuring that the product of the roots remains 33 or $-33$.

  Considering two distinct scenarios:

  **Case 1:** One of the roots is 33 or $-33$. In this scenario, the only other viable roots are $\pm 1$. Here, the root sum's absolute value is at least 32.

  **Case 2:** One root is 11 or $-11$, and another is 3 or $-3$. The only other plausible roots remain $\pm 1$, leading to a root sum's absolute value of at least $11 - 3 - 1 = 7$. This is a more optimal outcome than Case 1. For an absolute root sum of 7, we find $|a_{n-1}| = 7|a_n| = 14$.

  From our analysis, we deduce that $|a_{n-1}| \geq 14$. Verifying this, we observe that equality is attained by

$$f(x) = 2(x + 11)(x - 3)(x - 1)$$
$$= 2x^3 + 14x^2 - 82x + 66,$$

  which is consistent with having integer coefficients and integer roots. Thus, the minimal achievable value for $|a_{n-1}|$ is $\boxed{14}$.

- **Answer**: **14**

Figure 6: An example from the MATH dataset (Hendrycks et al., 2021; Lightman et al., 2023).

**[Chain-of-Thought Reasoning by GPT-4]**

- **Reasoning**: "Let's think step by step:
    1. The roots of the polynomial are distinct integers, and the leading coefficient is 2, so the roots must be divisors of $\frac{66}{2} = 33$.
    2. The divisors of 33 are $\pm 1, \pm 3, \pm 11, \pm 33$.
    3. The sum of the roots of the polynomial is $-\frac{a_{n-1}}{a_n}$.
    4. To minimize $|a_{n-1}|$, we should select the roots to be as close to zero as possible.
    5. The smallest possible sum of the roots is $-1 - 3 + 11 + 33 = 40$,
    6. Thus, the smallest possible value of $|a_{n-1}|$ is $2 \times 40 = \boxed{80}$."

- **Answer**: **80** (**Wrong**)

---

**[Cumulative Reasoning by GPT-4]**

- **Generated Hints**:
    1. "**Hint 1**: Recall Vieta's formulas, which relate the coefficients of a polynomial to the sums and products of its roots."
    2. "**Hint 2**: The product of the roots of the polynomial is equal to $-\frac{a_0}{a_n}$."
    3. "**Hint 3**: The sum of the roots of the polynomial is equal to $-\frac{a_{n-1}}{a_n}$."
    4. "**Hint 4**: Since the roots are distinct integers, consider the factors of $-\frac{a_0}{a_n}$ and their sums."

- **Generated Simple Questions and Answers**:

**Question 1:** "What is the product of the roots of the polynomial $f(x)$?"

    **Answer 1:** "The product of the roots of the polynomial is $-\frac{a_0}{a_n} = -\frac{66}{2} = -33$."

**Question 2:** "What are the possible sets of distinct integer roots of the polynomial?"

    **Answer 2:** "The possible sets of distinct integer roots are $\{-33, 1\}$, $\{-11, 3\}$, and $\{-3, 11, -1\}$."

**Question 3:** "What are the sums of the roots for each possible set of roots?"

    **Answer 3:** "The sums of the roots are $-32$, $-8$, and 7 for the sets $\{-33, 1\}$, $\{-11, 3\}$, and $\{-3, 11, -1\}$, respectively."

**Question 4:** "What is the least possible value of $|a_{n-1}|$?"

    **Answer 4:** "The least possible value of $|a_{n-1}|$ is $2 \times 7 = 14$."

- **Solution**: "Let's think step by step. Given that $a_n = 2$ and $a_0 = 66$, and the roots of the polynomial are distinct integers, the least possible value of $|a_{n-1}|$ is $\boxed{14}$."

- **Answer**: **14** (**Correct**)

---

Figure 7: Solutions for the example presented in Figure 6 from the MATH dataset, generated by CoT and CR. CoT will generate the answer directly through a chain of thought. By contrast, CR will first generate a few hints, then several simple and foundational questions, and then answer them by self, and finally conclude with the help of the generated hints and question-answer pairs.

# E More on Datasets

## E.1 More FOLIO Examples

---

[**Example ID: 603**]

- **Hypothesis**: Hyraxes lay eggs.

- **Premises**:

    1. The only types of mammals that lay eggs are platypuses and echidnas.
    2. Platypuses are not hyrax.
    3. Echidnas are not hyrax.
    4. No mammals are invertebrates.
    5. All animals are either vertebrates or invertebrates.
    6. Mammals are animals.
    7. Hyraxes are mammals.
    8. Grebes lay eggs.
    9. Grebes are not platypuses and also not echidnas.

- **Label**: [**False**]

---

[**Chain-of-Thought Reasoning by GPT-4**]

- **Reasoning**: "The premises state that the only types of mammals that lay eggs are platypuses and echidnas. Hyraxes are mammals but are neither platypuses nor echidnas. Since the conclusion is about hyraxes laying eggs but there's no direct information in the premises on this topic, the conclusion cannot be determined with certainty."

- **Prediction**: [**Unknown**] (**Wrong**)

---

[**Cumulative Reasoning by GPT-4**]

- **Generated Propositions**:

    1. "All mammals are either vertebrates or invertebrates."
    2. "Hyraxes do not lay eggs, as they are neither platypuses nor echidnas."

- **Reasoning**: "We can deduce that the only types of mammals that lay eggs are platypuses and echidnas. Hyraxes are mammals, but they are neither platypuses nor echidnas. Therefore, hyraxes do not lay eggs."

- **Prediction**: [**False**] (**Correct**)

---

[**Example ID: 546**]

- **Hypothesis**: Extractive models are not always faithful.

- **Premises**:

    1. A summarization model is always faithful if it uses content from the input documents.
    2. Extractive models are a kind of summarization models.
    3. Extractive model can only use content from the input documents.

- **Label**: [**False**]

---

[**Chain-of-Thought Reasoning by GPT-4**]

- **Reasoning**: "The premises state that a summarization model is always faithful if it uses content from the input documents. Extractive models are a kind of summarization model, and they can only use content from the input documents. However, the premises do not provide any information about whether extractive models are always faithful or not."

- **Prediction**: [**Unknown**] (**Wrong**)

[**Cumulative Reasoning Reasoning by GPT-4**]

- **Generated Propositions**:

  1. "All extractive models are faithful summarization models since they use content from the input documents."
  2. "All faithful summarization models use content from the input documents."

- **Reasoning**: "We can infer that extractive models are always faithful because they only use content from the input documents and satisfy the condition for a summarization model to be faithful. Therefore, the conclusion that extractive models are not always faithful is incorrect."

- **Prediction**: [**False**] (**Correct**)

[**Example ID: 404**]

- **Hypothesis**: No coursing is banned.

- **Premises**:

  1. Greyhound racing is a competitive sport where spectators bet on greyhounds.
  2. Coursing is a part of Greyhound racing.
  3. Some competitive sports where spectators bet on things are banned.
  4. Coursing involves spectators betting on a hare being pursued by greyhounds.
  5. Hares are small game.
  6. If a competitive sport involves spectators betting on small game, then it is banned.

- **Label**: [**False**]

[**Chain-of-Thought Reasoning by GPT-4**]

- **Reasoning**: "The premises state that coursing is a part of greyhound racing, and it involves spectators betting on a hare being pursued by greyhounds. Hares are small game, and if a competitive sport involves spectators betting on small game, then it is banned. It's also mentioned that some competitive sports where spectators bet on things are banned. However, the premises do not clearly state the status of coursing being banned or not banned."

- **Prediction**: [**Unknown**] (**Wrong**)

---

**[Cumulative Reasoning Reasoning by GPT-4]**

- **Generated Propositions**:

  1. "If a competitive sport involves spectators betting on hares, then it is banned."
  2. "Coursing is a competitive sport that involves spectators betting on small game."

- **Reasoning**: "We know that coursing is a competitive sport involving spectators betting on small game, specifically hares. The premises also state that if a competitive sport involves spectators betting on small game, then it is banned. Thus, we can deduce that there is a possibility that coursing is banned. The hypothesis claims that no coursing is banned, which contradicts the deduction that coursing might be banned."

- **Prediction**: [**False**] (**Correct**)

---

**[Example ID: 609]**

- **Hypothesis**: Allison has room for broccoli.

- **Premises**:

  1. If you have room for dessert, you have room for broccoli.
  2. Everyone at Luis's dinner party has room for dessert, including Luis.
  3. Mauricia does not have room for broccoli.
  4. Luis's dinner party is the first ever dinner party that Allison has attended.
  5. Gustave has room for both broccoli and asparagus.
  6. Broccoli and asparagus are both vegetables.

- **Label**: [**True**]

---

**[Chain-of-Thought Reasoning by GPT-4]**

- **Reasoning**: "The premises state that if you have room for dessert, you have room for broccoli. Everyone at Luis's dinner party has room for dessert, including Luis. However, we do not have information specifically about Allison having room for dessert or not, even though it's her first dinner party. Thus, we can't conclude with certainty if Allison has room for broccoli."

- **Prediction**: [**Unknown**] (**Wrong**)

---

**[Cumulative Reasoning by GPT-4]**

- **Generated Propositions**:

  1. "Gustave has room for vegetables."
  2. "Everyone at Luis's dinner party, including Mauricia, has room for dessert."

- **Reasoning**: "We know that:
  1. If one has room for dessert, they have room for broccoli.
  2. Everyone at Luis's dinner party, including Luis and Allison, has room for dessert.
  Using these premises, we can infer that since everyone at Luis's dinner party has room for dessert, they also have room for broccoli. Therefore, Allison, who is at Luis's dinner party, must have room for broccoli."

- **Prediction**: [**True**] (**Correct**)

---

### E.2 Curating FOLIO wiki dataset

1. Missing common knowledge or contradictory to common knowledge; (9 in total, Example ID No. 34, 62, 162, 167, 228, 268, 526, 677, 679)
2. Overly ambiguous problems failing to provide unequivocal answers; (37 in total, Example ID No. 141, 215, 216, 223, 252, 261, 298, 321, 330, 396, 402, 409, 411, 431, 432, 456, 457, 482, 483, 496, 563, 572, 599, 624, 629, 641, 654, 660, 673, 682, 698, 750)
3. Inherent inconsistencies presented within the premises; (2 in total, Example ID No. 640, 643)
4. Vague premises or typographical errors; (2 in total, Example ID No. 314, 315)
5. Incorrect answers. (24 in total, Example ID No. 9, 46, 52, 84, 100, 144, 273, 276, 299, 310, 322, 345, 367, 437, 452, 453, 464, 557, 573, 578, 605, 632, 671, 715)

---

[**Problem Description**]

- Example ID: 679
- **Premises**:
    1. Zaha Hadid is a British-Iraqi architect, artist and designer.
    2. Zaha Hadid was born on 31 October 1950 in Baghdad, Iraq.
    3. Hadid was a visiting professor of Architectural Design at the Yale School of Architecture.
    4. Max is an aspiring architecture student, and he plans to apply to Yale School of Architecture.
- **Hypothesis**: Hadid was born in 1982.
- **FOL Label**: [**Unknown**]
- **Human Label**: [**False**]
- **Explanation**: *We can see that Zaha Hadid was born on 31 October 1950 in Baghdad, Iraq. This directly contradicts the hypothesis that Hadid was born in 1982. It is common knowledge that people are born only once, and someone can't be born in two different years.*

---

Figure 8: Example 679 from the FOLIO wiki dataset, the origin label provided by the FOL system is not correct, so we choose to curate this dataset, removing these examples with wrong labels. For more examples, please refer to Appendix E.3.

### E.3 More examples on problems excluded from FOLIO wiki curated

**Type 1 Error: Missing common knowledge or contradictory to common knowledge**

---

[**Example ID: 34**]

- **Premises**:

  1. The Croton River watershed is the drainage basin of the Croton River.
  2. The Croton River is in southwestern New York.
  3. Kings are male.
  4. Water from the Croton River watershed flows to the Bronx.
  5. The Bronx is in New York.

- **Hypothesis**: Water from the Croton River flows to the Bronx.

- **Label**: [**Unknown**]

- **Wrong Type**: *[Type 1: Missing common knowledge or contradictory to common knowledge in the premises]*

- **Explanation**: *We understand that the Croton River is in southwestern New York, and the Bronx is also located in New York. It is stated that water from the Croton River watershed flows to the Bronx, and the Croton River watershed is the drainage basin of the Croton River. It is common knowledge that water from a river flows to its drainage basin. Therefore, it is true that water from the Croton River flows to the Bronx.*

---

[**Example ID: 268**]

- **Premises**:

  1. Bernarda Bryson Shahn was a painter and lithographer.
  2. Bernarda Bryson Shahn was born in Athens, Ohio.
  3. Bernarda Bryson Shahn was married to Ben Shahn.
  4. People born in Athens, Ohio are Americans.

- **Hypothesis**: Bernarda Bryson Shahn was born in Greece.

- **Label**: [**Unknown**]

- **Wrong Type**: *[Type 1: Missing common knowledge or contradictory to common knowledge in the premises]*

- **Explanation**: *We know that Bernarda Bryson Shahn was born in Athens, Ohio. It is common knowledge that Greece is not in Ohio. It also states that people born in Athens, Ohio, are Americans. Thus, it is false to conclude that Bernarda Bryson Shahn was born in Greece.*

---

**[Example ID: 62]**

- **Premises**:

    1. The Golden State Warriors are a team from San Francisco.
    2. The Golden State Warriors won the NBA finals.
    3. All teams attending the NBA finals have more than thirty years of history.
    4. Boston Celtics are a team that lost the NBA finals.
    5. If a team wins the NBA finals, then they will have more income.
    6. If a team wins or loses at the NBA finals, then they are attending the finals.

- **Hypothesis**: The Golden State Warriors will have more income for gate receipts.

- **Label**: [**True**]

- **Wrong Type**: *[Type 1: Missing common knowledge or contradictory to common knowledge in the premises]*

- **Explanation**: *We know that the Golden State Warriors won the NBA finals and that if a team wins the NBA finals, they will have more income. Therefore, we can infer that the Golden State Warriors will have more income. However, the hypothesis mentions 'more income for gate receipts,' and there is no information about gate receipts on the premises.*

---

**Type 2 Error: Overly ambiguous problems failing to provide unequivocal answers**

---

**[Example ID: 496]**

- **Premises**:

    1. Some fish may sting.
    2. Stonefish is a fish.
    3. It stings to step on a stonefish.
    4. Stonefish stings cause death if not treated.
    5. To treat stonefish stings, apply heat to the affected area or use an antivenom.

- **Hypothesis**: If you step on a stonefish and apply heat to the affected area, stings will cause death.

- **Label**: [**Unknown**]

- **Wrong Type**: *[Type 2: Overly ambiguous problems failing to provide unequivocal answers]*

- **Explanation**: *The premises state that applying heat to the affected area or using antivenom can treat stonefish stings. Thus, if heat is applied to the affected area, it should help treat the sting and prevent death. However, it is not certain that applying heat to the affected area will prevent death, as it is possible that the sting is too severe to be treated with heat.*

---

[**Example ID: 432**]

- **Premises**:

    1. Vic DiCara plays guitar and bass.
    2. The only style of music Vic DiCara plays is punk music.
    3. Vic DiCara played in the band Inside Out.

- **Hypothesis**: If you step on a stonefish and apply heat to the affected area, stings will cause death.

- **Label**: [**Unknown**]

- **Wrong Type**: *[Type 2: Overly ambiguous problems failing to provide unequivocal answers]*

- **Explanation**: *We know that Vic DiCara played in the band Inside Out and the only style of music he plays is punk music. This information implies that Inside Out played punk music while Vic DiCara was a member. However, it is not certain that Inside Out was a punk band, as it is possible that the band played a different style of music before Vic DiCara joined.*

[**Example ID: 673**]

- **Premises**:

    1. Cancer biology is finding genetic alterations that confer selective advantage to cancer cells.
    2. Cancer researchers have frequently ranked the importance of substitutions to cancer growth by P value.
    3. P values are thresholds for belief, not metrics of effect.

- **Hypothesis**: Cancer researchers tend to use the cancer effect size to determine the relative importance of the genetic alterations that confer selective advantage to cancer cells.

- **Label**: [**Unknown**]

- **Wrong Type**: *[Type 2: Overly ambiguous problems failing to provide unequivocal answers]*

- **Explanation**: *We can deduce that cancer researchers tend to use P values, not effect sizes, to rank the importance of genetic alterations. Thus, the hypothesis contradicts the premises. However, it is still possible that cancer researchers use the cancer effect size to determine the relative importance of the genetic alterations that confer selective advantage to cancer cells.*

**Type 3 Error: Inherent inconsistencies presented within the premises**

[**Example ID: 640**]

- **Premises**:

    1. William Dickinson was a British politician who sat in the House of Commons.
    2. William Dickinson attended Westminster school for high school and then the University of Edinburgh.
    3. The University of Edinburgh is a university located in the United Kingdom.
    4. William Dickinson supported the Portland Whigs.
    5. People who supported the Portland Whigs did not get a seat in the Parliament.

- **Hypothesis**: William Dickinson did not get a seat in the Parliament.

- **Label**: [**True**]

- **Wrong Type**: *[Type 3: Inherent inconsistencies presented within the premises]*

- **Explanation**: *We have a contradiction. On one hand, we have information that William Dickinson supported the Portland Whigs, and people who supported the Portland Whigs did not get a seat in the Parliament. On the other hand, another premise states that William Dickinson was a British politician who sat in the House of Commons, which implies that he did get a seat in the Parliament.*

[**Example ID: 643**]

- **Premises**:

    1. William Dickinson was a British politician who sat in the House of Commons.
    2. William Dickinson attended Westminster school for high school and then the University of Edinburgh.
    3. The University of Edinburgh is a university located in the United Kingdom.
    4. William Dickinson supported the Portland Whigs.
    5. People who supported the Portland Whigs did not get a seat in the Parliament.

- **Hypothesis**: William Dickinson sat in the House of Commons.

- **Label**: [**True**]

- **Wrong Type**: *[Type 3: Inherent inconsistencies presented within the premises]*

- **Explanation**: *We have a contradiction. On one hand, we have information that William Dickinson supported the Portland Whigs, and people who supported the Portland Whigs did not get a seat in the Parliament. On the other hand, another premise states that William Dickinson was a British politician who sat in the House of Commons, which implies that he did get a seat in the Parliament.*

**Type 4 Error: Vague premises or typographical errors**

[**Example ID: 314**]

- **Premises**:

    1. Palstaves are a type of early bronze axe.
    2. Commonly found in northern, western and south-western Europe, palstaves are cast in moulds.
    3. John Evans is an archeologist who popularized the term "palstave".
    4. A paalstab is not an axe, but rather a digging shovel.

- **Hypothesis**: John Evans Popularized the term paalstab.

- **Label**: [**Unknown**]

- **Wrong Type**: *[Type 4: Vague premises or typographical errors]*

- **Explanation**: *What is palstave and paalstab? Were they misspelled?*

---

[**Example ID: 315**]

- **Premises**:

    1. Palstaves are a type of early bronze axe.
    2. Commonly found in northern, western and south-western Europe, palstaves are cast in moulds.
    3. John Evans is an archeologist who popularized the term "palstave".
    4. A paalstab is not an axe, but rather a digging shovel.

- **Hypothesis**: There is an axe that is commonly found in Western Europe.

- **Label**: [**Unknown**]

- **Wrong Type**: *[Type 4: Vague premises or typographical errors]*

- **Explanation**: *We can see that palstaves are a type of early bronze axe and they are commonly found in northern, western, and south-western Europe. Therefore, it is true that there is an axe that is commonly found in Western Europe. However, the premises also state that a paalstab is not an axe, but rather a digging shovel. Was paalstab the same thing as palstaves?*

**Type 5 Error: Incorrect answers**

[**Example ID: 9**]

- **Premises**:

    1. Palstaves are a type of early bronze axe.
    2. Pierre de Rigaud de Vaudreuil built Fort Carillon.
    3. Fort Carillon was located in New France.
    4. New France is not in Europe.

- **Hypothesis**: Fort Carillon was located in Europe.

- **Label**: [**Unknown**]

- **Wrong Type**: *[Type 5: Incorrect answers]*

- **Explanation**: *We know that Fort Carillon was located in New France, and New France is not in Europe. Therefore, Fort Carillon was not located in Europe.*

[**Example ID: 632**]

- **Premises**:

    1. New York City is on the East Coast.
    2. Seattle is on the West Coast.
    3. If a person from a city on the East coast is traveling to a city on the west coast, they will be on a long flight.
    4. Most passengers on flights to Seattle from New York City are not in first class.
    5. People on long flights are uncomfortable unless they're in first class.

- **Hypothesis**: Some people flying from New York City to Seattle will be uncomfortable.

- **Label**: [**False**]

- **Wrong Type**: *[Type 5: Incorrect answers]*

- **Explanation**: *We can deduce the following: 1. A person traveling from New York City to Seattle will be on a long flight (since New York City is on the East Coast and Seattle is on the West Coast). 2. Most passengers on flights from New York City to Seattle are not in first class. 3. People on long flights are uncomfortable unless they're in first class. Given this information, we can conclude that some people flying from New York City to Seattle will be uncomfortable, as most of them are not in first class and long flights cause discomfort for those not in first class.*

---

[**Example ID: 671**]

- **Premises**:

    1. Westworld is an American science fiction-thriller TV series.
    2. In 2016, a new television series named Westworld debuted on HBO.
    3. The TV series Westworld is adapted from the original film in 1973, which was written and directed by Michael Crichton.
    4. The 1973 film Westworld is about robots that malfunction and begin killing the human visitors.

- **Hypothesis**: Michael Crichton has directed a film about robots.

- **Label**: [**Unknown**]

- **Wrong Type**: *[Type 5: Incorrect answers]*

- **Explanation**: *We can deduce that Michael Crichton wrote and directed the 1973 film Westworld, which is about robots that malfunction and begin killing the human visitors. Thus, it is true that Michael Crichton has directed a film about robots.*

## F   Appendix for Prompts

The design of few-shot prompts is critical to guiding the behavior of each LLM role within CR. We crafted these prompts intending to encapsulate the essence of each role:

- The Proposer prompt encourages the generation of plausible next steps or hypotheses.

- The Verifier prompt focuses on assessing the validity of these propositions.

- The Reporter prompt aims at determining the sufficiency of information for concluding the reasoning process.

There have been several works (Reynolds & McDonell, 2021; Zhang et al., 2024a; Zhou et al., 2024) showing that zero-shot meta-prompts can also work well, which minimizes the bias introduced in the few-shot examples.

---

**System**: Suppose you are one of the greatest AI scientists, logicians, and mathematicians. Let us think step by step. Please use First-Order Logic (FOL) to deduce a "Proposition" from two given "Premises". Please make sure that the "Proposition" is logically correct. Please make sure that the "Proposition" is not a duplicate of the "Premises". Please make sure your reasoning is directly deduced from the "Premises" and "Propositions" rather than introducing unsourced common knowledge and unsourced information by common sense reasoning. Please remember that your "Proposition" should be useful to determine whether the Hypothesis is True, False, or Unknown.

**User**:
"Premises": "{this.premises}"
We want to deduce more propositions to determine the correctness of the following Hypothesis:
"Hypothesis": "{this.hypothesis}"

**Assistant**:
"Proposition": "{[to be generated]}"

---

Figure 9: Prompt template for CR Proposer on logical inference tasks.

---

**System**: Suppose you are one of the greatest AI scientists, logicians, and mathematicians. Let us think step by step. Please use First-Order Logic (FOL) to determine whether the deduction of two given "Premises" to a "Proposition" is valid or not, and reply with True or False.

**User**:
"Premises": "{this.premises}"
"Proposition": "{this.proposition}"

**Assistant**:
"Judgement": "Is this deduction valid? {[True or False]}"

---

Figure 10: Prompt template for CR Verifier on logical inference tasks.

**System**: Suppose you are one of the greatest AI scientists, logicians, and mathematicians. Let us think step by step. Read and analyze the "Premises" first, then use First-Order Logic (FOL) to judge whether the "Hypothesis" is True, False, or Unknown. Please make sure your reasoning is directly deduced from the "Premises" and "Propositions" rather than introducing unsourced common knowledge and unsourced information by common sense reasoning.

**User**:
"Premises": "{this.premises}"
"Hypothesis": "{this.hypothesis}"

**Assistant**:
"Thoughts": "Let us think step by step. From the premises, we can deduce propositions: {this.propositions}"
"Recall the Hypothesis": "{[this.hypothesis}"
"Judgement": "Now we know that the Hypothesis is {[True or False]}"

Figure 11: Prompt template for CR Reporter on logical inference tasks.

**System**: Suppose you are one of the greatest AI scientists, logicians, and mathematicians. You are very good at basic arithmetic operations. Use numbers and basic arithmetic operations (+ - * /) to obtain 24 with input numbers. In each step, You are only allowed to randomly choose arbitrary TWO of the input numbers to obtain a new number using arbitrary one basic arithmetic operation (AVOID duplicating with forbidden steps). Your calculation process must be correct.

**User**: Input: [a, b, c, d]
Next Step:

**Assistant**: c * d = e

**User**: Remaining Numbers:

**Assistant**: [a, b, e]

Figure 12: Prompt template for CR Proposer on Game of 24.

**System**: Suppose you are one of the greatest AI scientists, logicians, and mathematicians. You are very good at basic arithmetic operations. Use numbers and basic arithmetic operations (+ - * /) to obtain 24 with input numbers. Evaluate if the given intermediate step is correct and only use two existing numbers.

**User**:
Input: 10, 14
Intermediate step: 10 + 14 = 24

**Assistant**:
The intermediate step is valid.
Judgement:
Valid

**User**: Input: [a, b]
Intermediate step: [a op b = result]

**Assistant**:
{[reasoning to be generated]}
Judgement:
{[Valid or Invalid]}

Figure 13: Prompt template for CR Verifier (a) on Game of 24.

**System**: Suppose you are one of the greatest AI scientists, logicians, and mathematicians. You are very good at basic arithmetic operations. Use numbers and basic arithmetic operations (+ - * /) to obtain 24 with input numbers. Evaluate if given numbers can reach 24 (sure/likely/impossible).

**User**:
Input: 10, 14
Draft:

**Assistant**:
14 - 10 = 4
14 * 10 = 140
10 / 14 = 5/7
14 / 10 = 1.4
10 + 14 = 24

**User**:
Input: {remaining_numbers}
Draft:

**Assistant**:
sure
10 + 14 = 24

**User**:
{[reasoning to be generated]}
Judgement:
{[Valid or Invalid]}

Figure 14: Prompt template for CR Verifier (b) on Game of 24.

**System**: Suppose you are one of the greatest AI scientists, logicians, and mathematicians. You are very good at basic arithmetic operations. Use numbers and basic arithmetic operations (+ - * /) to obtain 24 with input numbers. You need to combine the given intermediate steps step-by-step into a complete expression.

**User**:
Input: 1, 1, 4, 6
Intermediate steps:
1 * 4 = 4 (left 1, 4, 6)
1 * 4 * 6 = 24

**Assistant**:
Draft:
Because 1 * 4 * 6 = 24, while 1 * 4 = 4. So 1 * (1 * 4) * 6 = 24.
Output:
1 * (1 * 4) * 6 = 24

**User**:
Input: {input}
Intermediate steps:
{intermediate steps}

**Assistant**:
Draft:
{[to be generated]}
Output:
{[to be generated]}

Figure 15: Prompt template for CR Reporter on Game of 24.

```
<syntax>

## Problem: [problem]

Solution: Lets' think step by step. [somewords interpreting the origin problem]

### Preliminary Contents

- **Prelim 1**: [preliminary contents 1]

- **Prelim 2**: [preliminary contents 2]

- [...]

### Hints
- **Hint 1**: [useful hints 1]

- **Hint 2**: [useful hints 2]

- [...]

### Intermediate Steps: Question-AnswerSketch-Code-Output-Answer Pairs

Let's think step by step.

#### Question 1: [the first question you raised]
- **Answer Sketch**: [write a sketch of your answer to question 1]

##### Code for Question 1
[call code interpreter here to verify and solve your answer sketch to question 1]

#### Answer for Question 1
- **Answer**: [your answer to this question 1 based on the results
given by code interpreter (if presented)]

#### Question 2: [the second question you raised]
- **Answer Sketch**: [write a sketch of your answer to question 2]

##### Code for Question 2
[call code interpreter here to verify and solve your answer sketch to question 2]

#### Answer for Question 2
- **Answer**: [your answer to this question 2 based on the results
given by code interpreter (if presented)]

#### Question 3: [the third question you raised]
- **Answer Sketch**: [write a sketch of your answer to question 3]

##### Code for Question 3
[call code interpreter here to verify and solve your answer sketch to question 3]

#### Answer for Question 3
- **Answer**: [your answer to this question 3 based on the results
given by code interpreter (if presented)]

### [Question ...]

### Final Solution:

Recall the origin problem <MathP> [origin problem] </MathP>.

Let's think step by step.

#### Solution Sketch
[write a sketch for your final solution]

#### Code for Final Solution
[call code interpreter here to verify and solve your final solution]

#### Final Answer
[present the final answer in latex boxed format, e.g., $\boxed{63\pi}$]
Final Answer: the answer is $\boxed{...}$.

</syntax>
---
```

Figure 16: Meta Prompt for CR with code environment on solving MATH problems.

