# OpenReview forum: "Cumulative Reasoning with Large Language Models"
_TMLR — Accepted by TMLR_

### Review · Reviewer_vNxH · 2025-04-15

**Summary Of Contributions:**

This work introduces a new prompting technique called “Cumulative Reasoning” (CR). The idea is that it should decompose a reasoning task into smaller tasks, and re-use previous “verified” propositions to compose them to enhance the problem-solving capabilities of LLMs.

The idea is based on the fact that the reasoning logical steps are better represented by a direct acyclic graph (DAG) instead of a linear sequence of deductions. In order to enforce a DAG structure in the LLM “steps of thought” they have three subprocesses: the proposer, the verifier, and the reporter.

**Audience:**

No

**Broader Impact Concerns:**

No concerns.

**Claims And Evidence:**

No

**Requested Changes:**

All my concerns must be tackled for me to consider the work to be ready for publication. The only exception is the last point on comparing with more modern LLMs because I do realise that it is not always easy to access the most recent models, even though I believe an effort into that direction would make the work way more solid, future-proof, and interesting for the community. At the very least, however, it is necessary to refresh the related works section.

**Strengths And Weaknesses:**

Overall, the idea is interesting and aligned with the recent trends in the LLM research that a proposer-verifier framework leads to improved results and better consistency.

The paper is also generally well written and easy to follow. It is undeniable that with their technique the authors got interesting improvements compared with the baselines they use. This can give some insights on how to effectively prompt LLMs to improve their performance on reasoning tasks.

However, there are several things that I find problematic (my feedback might sound harsh but I really intend it to be polite and constructive, I hope this is getting through it). Specifically:
- The method section (Section 3) is just saying that they split the LLM prompt in three parts, then all the details are in the appendix. Moreover, the way it is described is very redundant and fuzzy. The authors refer to “intuitionistic logic” and “the philosophy of mathematical constructivism” without any reference on why these fields are making the difference for their approach compared to the baselines. A lot of it sounds like “empty words” that sounds convincing but they are not technically contributing to the method definition.
- The verifier is defined as the part that “translate(s) the candidate steps into formal representations—using symbolic reasoning systems or an integrated code environment—to ensure logical soundness” and then in section 3.1 they “…assume that whenever a verifier is employed, its accuracy is near-perfect”. But then in practice they use just another prompt to LLMs to do the verification part, and that for sure does not ensure any logical soundness, nor satisfy the near-perfect assumption. Only for one experiment they use a real verifier. That, I believe, is the most interesting part of the paper. From the definition of the method it looked like the entire paper was focused on this (especially considering the preliminaries on logic, and the appendix on higher order logic). I highly recommend rephrasing the misleading parts, or focus on actually using reliable verifiers.
- Section 3.1 is entitled “Comparison with CoT and ToT” but it is not really a comparison. One would expect to finish reading the section with a clear idea of how CR differs with respect to CoT and ToT, but that is not the case. The authors have a nice figure (Fig. 3) that showcases it, but they do not describe it. Many details are left to the reader to figure out (and the text is very small). For example, the distribution over the previous nodes seems to be a key aspect (they condition each step on all the previous steps, instead of keeping only “linear” dependencies), but that is not discussed at all. This also seems to increase the complexity of the method (adding variables to the conditioning set can increase exponentially the modelled distribution).
- Moreover, still in Section 3.1, the entire point is to “prove” that CR gives correct results with higher probability with respect to CoT/ToT. The problem is that Assumption 3.2 is very strong, and once that is assumed, Theorem 3.4 is trivially satisfied. The assumption (please correct me if I’m wrong) is saying that a single step of CR is more likely to lead to a correct result than ToT, and, what is even more problematic, is that this is a monotonic property that keeps holding when new nodes are added. The authors say that this assumption has been empirically validated in various tasks but as far as I understand it: (1) they should verify that p_ToT <= p_CR, and that cannot be done by related works since *this* is the work introducing CR; (2) they would need to show that at *every* step the assumption holds, and I do not see experiments showing this. It is in fact probably not hard to see that this assumption does not hold by analysing LLMs output traces: in practice, refinements can lead to poorer results because of hallucinations or other statistical artefacts, no? Earlier in the section, they also “assume that a unique correct reasoning path exists for the problem”. Why? Isn’t this quite limiting?
- Finally, one might wonder what are the real insights provided by the CR method. As I said, the improvements are undeniable wrt to the baselines used, but the technique of using a proposer-verifier is not really new [1,2], and the authors compare CR only with CoT/ToT. So what are the insights for the scientific community? Moreover, there are works showing the computational limitations of autoregressive models [2,3], and CR seems to provide an advantage because it trades time and memory for accuracy, like other similar works, or the most recent models do. Is this the case? What would be interesting is showing how efficient this method is compared to other similar ones in terms of tokens used, for example. If a DAG-like structure is exploited to keep the reasoning dependencies under control that could be a neat advantage.
- It is also worth noticing that the models used in this work are quite outdated now, and one might wonder if the more recent models that are trained to mimic reasoning (eg. DeepSeek R1) already solve this problems more efficiently than using CR..?

[1] Xiang et al. “Towards System 2 Reasoning in LLMs: Learning How to Think With Meta Chain-of-Thought”
[2] Ahn et al. “Do As I Can, Not As I Say: Grounding Language in Robotic Affordances”
[3] Peng et al. “On Limitations of the Transformer Architecture”
[4] Hazra et al. “Can large language models reason? A characterisation via 3-SAT”

---

> ### Author Response · Authors · 2025-04-19
> **Rebuttal Comment Part I**
>
> Dear Reviewer vNxH, Editors,
>
> Thank you for your thorough review and constructive feedback on our manuscript. We appreciate your positive comments on the interestingness of the core idea, the clarity of the writing, and the significance of the empirical improvements demonstrated by Cumulative Reasoning (CR). We value the time and effort you invested in assessing our work.
>
> We have carefully considered all the points you raised and offer the following responses and planned revisions. We hope this clarifies our methodology and contributions.
>
> > Q1: The method section (Section 3) is just saying that they split the LLM prompt into three parts, ...
>
> A1: We understand your concern that Section 3 provides a high-level overview, while detailed prompts are in the appendix. This was a deliberate choice to maintain the flow of the main paper, focusing on the conceptual framework of CR (Proposer, Verifier, Reporter roles) and its iterative, cumulative nature[cite: 1]. We will revise Section 3 to integrate slightly more detail about the role interactions and prompt design principles, guiding the reader more effectively to the specific implementations detailed in Appendix F.
>
> We apologize if the references to intuitionistic logic and mathematical constructivism seemed redundant. Our intent was to ground CR's iterative, verification-driven approach in established paradigms that emphasize constructive proof and step-by-step validation, contrasting with methods that might explore less validated paths[cite: 1]. We will revise this section to connect these concepts to CR's design explicitly, clarifying *why* this philosophical grounding is relevant (e.g., emphasizing the construction of the solution DAG through verified steps) and adding appropriate citations if needed.
>
> > Q2: The verifier is defined as the part that “translate(s) the candidate steps into formal representations—using symbolic reasoning systems or an integrated code environment ...
>
> A2: We acknowledge the distinction you highlighted between the ideal definition of the verifier (ensuring logical soundness, near-perfect accuracy) and its practical implementation using LLM prompts in most experiments. The "near-perfect" assumption was primarily stated for the theoretical comparison in Section 3.1 and is more closely approximated when using symbolic verifiers like a code interpreter, as done in one set of experiments (MATH with code environment).
>
> Using LLMs as verifiers was a practical choice to demonstrate CR's applicability even without specialized symbolic tools across diverse tasks. While an LLM verifier doesn't guarantee formal logical soundness like a symbolic system, it serves as a crucial component for error checking and refinement within the CR framework, as supported by ablation studies (Table 5 shows a drop in accuracy without the verifier).
>
> We will rephrase the description of the verifier in Section 3 to more clearly distinguish between the conceptual role (validation) and the specific implementations (LLM-based vs. symbolic/code-based). We will explicitly state the limitations of using an LLM as a verifier while highlighting its practical benefits within the CR process. We will emphasize the MATH experiment results with the code environment, as demonstrating the potential when a more rigorous verifier is employed.
>
> > Q3: Section 3.1 is entitled “Comparison with CoT and ToT” but it is not really a comparison ...
>
> A3: Thank you for pointing out that the comparison in Section 3.1 could be clearer and that Figure 3 was underutilized. We will significantly revise Section 3.1. We will explicitly use Figure 3 to detail the structural differences: CoT's linearity, ToT's tree structure, and CR's DAG structure. We will elaborate on how CR's ability to condition on *all* previously validated steps (nodes in the DAG) provides a richer context compared to ToT's potentially more limited backtracking or CoT's linear history. We will also add a discussion on the potential complexity trade-offs associated with this richer context.

---

> ### Author Response · Authors · 2025-04-19
> **Rebuttal Comment Part II**
>
> > Q4. Regarding Assumption 3.2 and Theorem 3.4:
>
> A4: We acknowledge that Assumption 3.2 is strong. Its purpose was to formalize the intuition that verification (in ToT and CR) prevents exploring incorrect paths, and that CR's richer context (conditioning on more validated steps) makes generating the *next* correct step more likely than ToT. The assumption $p_{ToT} \le p_{CR}$ is based on CR leveraging a potentially larger set of verified intermediate results. While we cited empirical validation for related self-correction/refinement ideas, we agree that directly validating $p_{ToT} \le p_{CR}$ at every step for CR is complex and wasn't explicitly shown.
>
> We agree that Theorem 3.4 follows directly once Assumption 3.2 is accepted. The goal was to formally illustrate a potential theoretical advantage under these assumptions.
>
> The assumption of a unique correct path was made for simplifying the theoretical analysis (Definition 3.1), particularly for probability calculations. We recognize this is limiting and doesn't hold for all problems.
>
> We will re-evaluate the phrasing and justification for Assumption 3.2. We will clarify that it's a simplifying assumption for the theoretical model and better connect it to the empirical results where CR outperforms ToT (e.g., Game of 24, Logic Tasks). We will explicitly state the limitations of the unique path assumption. We will also add a brief discussion acknowledging that practical LLM refinements aren't always monotonic due to factors like hallucination, but argue the verifier's role in CR mitigates this overall.
>
> > Q5. Regarding Novelty, Baselines, Efficiency, and Insights:
>
> A5: While proposer-verifier frameworks exist, we believe CR's specific contribution lies in the *cumulative* and *structured* (DAG-based) way it builds upon verified intermediate steps, orchestrated by distinct Proposer, Verifier, and Reporter roles. This contrasts with linear chains (CoT) or fixed trees (ToT). The key insight is that explicitly managing and reusing a growing set of verified propositions within a flexible DAG structure enhances complex reasoning.
>
> CR's DAG approach, while potentially more complex contextually, might offer efficiencies by avoiding redundant exploration of invalid paths compared to broad searches. Table 7 shows CR uses significantly fewer visited states than ToT in Game of 24[cite: 1], and Appendix B provides some comparisons on visited states for logic tasks[cite: 1].
>
> We will refine the introduction and discussion to more clearly articulate the specific novelties of CR's cumulative DAG approach compared to prior work. We will expand the Related Work section (Section 5) to include and differentiate CR from the works you cited [1, 2] and other relevant proposer-verifier or iterative refinement methods. We will add a paragraph discussing computational considerations, referencing Appendix B and Table 7, while potentially noting a full efficiency benchmark across all tasks as an area for future work.
>
> > Q6: Regarding Model Versions and Related Work ...
>
> A6: We understand the concern about using models that might be considered outdated. As you noted, access to the very latest models can be challenging. We fully agree that the Related Work section needs refreshing. We will update Section 5 to include more recent relevant works, including those focusing on reasoning in newer models and potentially discussing how CR might complement or compare to their capabilities, even if direct empirical comparison isn't feasible for us currently.
>
> ---
>
> We believe addressing these points will significantly strengthen the paper. We appreciate the reviewer's constructive approach and hope our planned revisions will satisfy the concerns raised.
>
> Thank you again for your valuable feedback.
>
> Sincerely,
>
> The Authors

---

> > ### Comment · Reviewer_vNxH · 2025-04-22
> >
> > If you are able to tackle all the points, I do agree the work would improve a lot.
> >
> > I am still not entirely sure about the assumption (and following theorem), because you assume something that then cannot be ensured practically. But depending on how you phrase things, it might make the paper more solid, I suppose.
> >
> > Finally, I would indeed focus more on the DAG structure idea and on the formal verification of steps, that would be a clear difference even compared to the modern LRMs (other than the rest of the details you mentioned).
> >
> > One last point that I think it is worth to be mentioned is that this DAG structure is a complete DAG, so every node points to all other nodes (without creating cycles). So, for example, Figure 2 is misleading, I think. That should be clarified. And, personally, it would be interesting if you can optimise the DAG such that the nodes don't depend on *all* the previous nodes. Then you could compensate the computational complexity a bit. It is not clear to me now if you can already do it, but you decide not to because the accuracy gain is worth the computational cost, or if it is not possible at all.
> >
> > Good luck, and thank you for the polite answer.

---

> ### Author Response · Authors · 2025-04-22
>
> Dear Reviewer vNxH,
>
> Thank you very much for your prompt and thoughtful follow-up comments. We appreciate your engaging further with our work and providing additional constructive points.
>
> We acknowledge your remaining concerns regarding Assumption 3.2 and Theorem 3.4. We understand the skepticism about assuming properties that are difficult to guarantee perfectly in practice, especially when using LLMs as verifiers. In our revision, we will carefully rephrase this section to delineate the theoretical motivation from the practical implementation, ensuring we accurately represent the scope and limitations of the assumption and the theorem.
>
> We agree with your suggestion to place stronger emphasis on the unique aspects of CR, particularly the cumulative DAG structure and the integration of verification (especially the formal verification demonstrated with the code environment). We will ensure these aspects are highlighted more prominently in the revised manuscript as key differentiators from existing methods, including CoT, ToT, and potentially newer large reasoning models (LRMs).
>
> Regarding your insightful point on the DAG structure:
>
> 1. Our current implementation, as described conceptually and implied by the theoretical analysis, generally aims to leverage the context from *all* previously validated propositions within the reasoning process leading to a new candidate step. This is intended to provide the Proposer LLM with the richest possible context to generate the next step accurately.
> 2. We acknowledge that Figure 2 is a simplified illustration. It depicts dependencies within a particular successful reasoning path, but might not fully capture the notion of conditioning on *all* globally available validated nodes if multiple branches were explored. We will clarify this in the text and revise the figure caption or the figure itself to avoid misleading interpretations.
> 3. Your suggestion to potentially optimize the DAG dependencies (i.e., making it sparser by not conditioning on all previous nodes) is very interesting. Currently, our method prioritizes leveraging the full validated context to maximize accuracy, accepting the associated computational cost. We haven't systematically explored optimizing these dependencies, for example, by selecting only the most relevant prior steps. Determining relevance automatically and understanding the trade-off between context richness, computational cost, and accuracy is a valuable direction for future research. We will briefly note this potential optimization and trade-off in the revised version.
>
> Thank you again for your detailed feedback and guidance. We believe incorporating these clarifications will substantially improve the paper's clarity and contribution. We appreciate your time and constructive criticism.
>
> Sincerely,
>
> The Authors

---

### Review · Reviewer_zVDj · 2025-05-06

**Summary Of Contributions:**

The paper introduces Cumulative Reasoning (CR), a novel framework designed to enhance the reasoning capabilities of large language models (LLMs) by decomposing complex problems into smaller, manageable components and iteratively refining solutions through three specialized LLM roles: Proposer, Verifier, and Reporter. Inspired by Kahneman’s dual-process theory, CR shifts LLMs from fast, intuitive processing to more deliberate, stepwise reasoning. The method demonstrates strong empirical performance on several benchmarks, outperforming existing approaches like Chain-of-Thought (CoT) and Tree-of-Thought (ToT).

**Audience:**

Yes

**Broader Impact Concerns:**

No.

**Claims And Evidence:**

Yes

**Requested Changes:**

1. Currently, the innovativeness of  the proposed method is weak compared with other wokers.
2. More reasoner frameworks have been proposed, it is better to make some comparisions.

**Strengths And Weaknesses:**

Strengths:

1. CR introduces a structured, iterative approach that mimics human-like reasoning by dynamically storing and validating intermediate results, addressing a key limitation in prior methods like CoT and ToT.

2. The method achieves state-of-the-art (SOTA) performance across multiple benchmarks, including 98% accuracy in 24-game solving and 42% relative improvement in MATH Level 5 problems, demonstrating broad applicability.

3. By decomposing problems into smaller steps and validating intermediate results, CR better emulates human System 2 reasoning compared to single-pass or tree-based approaches.

Weaknesses:

1. Lack of Comparison with Latest Reasoning Methods: The paper does not benchmark CR against more recent reasoning frameworks such as Graph-of-Thought (GoT), Self-Consistency with Voting (SCV), Forest of Throught (FoT), etc., which could provide a clearer picture of its relative advantages.

2. Employing three separate LLM roles (Proposer, Verifier, Reporter) may increase computational costs compared to simpler prompting strategies (e.g., CoT-SC).

---

> ### Author Response · Authors · 2025-05-08
> **Rebuttal Comment Part I**
>
> Dear Reviewer, we sincerely thank your thorough review and insightful comments on our manuscript. We appreciate the positive feedback on the structured, iterative nature of CR, its emulation of human-like reasoning, and its strong empirical performance on various benchmarks. We will address the identified weaknesses and requested changes.
>
> > Lack of Comparison with Latest Reasoning Methods: The paper does not benchmark CR against more recent reasoning frameworks such as Graph-of-Thought (GoT), ...
>
> Thank you for your valuable suggestions regarding the comparison of CR with more recent reasoning frameworks like Graph-of-Thought (GoT), Self-Consistency with Voting (SCV), and Forest of Thought (FoT). We also appreciate your feedback concerning the innovativeness of CR in light of these and other existing works.
>
> We respectfully submit that CR offers distinct innovations. While Graph-of-Thought is a conceptually related area, CR operationalizes graph-based reasoning in a specific, structured way through its Proposer-Verifier-Reporter architecture. Indeed, any rational, multi-step logical reasoning process that leverages diverse intermediate conclusions can be seen as forming a Directed Acyclic Graph (DAG). Our work embraces this view; Section 2.2 ("Illustrative Example") explicitly models a derivation as a DAG (illustrated in Figure 1), and Section 3.1 ("Comparison with CoT and ToT") further details how CR organizes validated steps into such a structure. CR’s particular contribution lies in its explicit, iterative method for constructing and validating this DAG. The Proposer generates potential nodes and edges, the Verifier(s) stringently ensure their soundness before inclusion, and the Reporter synthesizes the final path from this verified graph. This clear division of labor, coupled with the cumulative utilization of all previously verified nodes to inform new proposals, forms the core of CR's unique approach. While GoT frameworks may also construct graph structures, CR’s emphasis on these specialized roles and the dynamic accumulation of evidence within a single, evolving DAG provides a specific and effective methodology.
>
> Our existing comparisons include CoT-Self-Consistency (CoT-SC), which shares fundamental principles with SCV, such as generating multiple solutions and selecting the most consistent one, often via majority voting. As shown in our results (e.g., Table 2, Table 3, Table 6), CR consistently outperforms CoT-SC. A key distinction is CR's iterative integration of verification within the reasoning process via the Verifier role, allowing for early pruning of erroneous paths, whereas SCV typically applies voting as a post-hoc selection over completed chains. Although CR's Reporter can incorporate voting, its primary strength derives from this intermediate verification. Similarly, Forest of Thought (FoT) methodologies often explore multiple diverse reasoning trees or high-level plans concurrently. CR, while capable of exploring branches (e.g., parameter `b` in our Game of 24 experiments in Table 7), primarily focuses on the meticulous, cumulative construction of a single, coherent reasoning DAG, rather than managing multiple, potentially independent, reasoning structures.
>
> To address these points more directly in the manuscript, we will revise Section 5 (Related Work) and Section 3.1 (Comparison with CoT and ToT). These revisions will explicitly discuss the conceptual relationships and distinctions between CR and frameworks like GoT, SCV, and FoT. We will further underscore how CR's Proposer-Verifier-Reporter architecture and its emphasis on the cumulative reuse of verified intermediate steps contribute to its novelty and effectiveness. While incorporating new, extensive experiments against all these recent methods is challenging for the current revision cycle due to resource constraints, our textual clarifications will aim to firmly situate CR within this advanced landscape, highlighting its unique contributions.

---

> ### Author Response · Authors · 2025-05-08
> **Rebuttal Comment Part II**
>
> > Employing three separate LLM roles (Proposer, Verifier, Reporter) may increase computational costs
>
> Thank you for your feedback on the computational costs of CR. You've raised a valid point that employing three LLM roles could potentially increase computational overhead compared to simpler prompting strategies.
>
> We acknowledge that CR, in its complete form, can involve more LLM interactions. This is a deliberate design consideration, reflecting a trade-off where increased computation is exchanged for significant improvements in accuracy and robustness. This benefit is especially pronounced on complex, multi-step problems where simpler methods often struggle, as evidenced by our results on FOLIO, Game of 24, and the challenging Level 5 MATH problems (detailed in Table 9, 10, 11 and 12).
>
> Several factors mitigate these costs. As noted in Section 4 (Experiments), all three roles (Proposer, Verifier, Reporter) are instantiated using the same underlying LLM, distinguished only by role-specific few-shot prompts, thereby avoiding the overhead of loading multiple distinct large models. Furthermore, the Verifier role is not restricted to an LLM; our MATH experiments with a code environment (Section 4.5, Tables 11 and 12) successfully employ a Python interpreter as a symbolic verifier, which is far less computationally demanding than an LLM call. This hybrid capability is a strength. It's also worth noting that in tasks like the Game of 24 (Table 7), CR achieved superior accuracy while exploring significantly fewer states than ToT, suggesting that its guided and verified approach can lead to greater search efficiency. Finally, parameters such as the number of intermediate propositions (`n` in FOLIO) or iteration limits can be adjusted to manage the balance between performance and computational budget.
>
> In our revised manuscript, we will add a discussion, likely in Section 6 (Conclusion) or a dedicated subsection, to explicitly address these computational considerations. This discussion will acknowledge the potential for increased inference calls while also contextualizing these with the substantial performance gains, the single-LLM architecture for roles, the utility of non-LLM verifiers, and instances of enhanced search efficiency. We will also suggest that optimizing the computational footprint of CR represents a valuable avenue for future research.
>
> We believe these proposed revisions will comprehensively address your concerns, further clarify CR's unique contributions and its position relative to other methods, and offer a balanced view of its computational profile. We are confident these changes will substantially strengthen our manuscript.
>
> Thank you once again for your valuable feedback.
>
> Sincerely,

---

> > ### Comment · Reviewer_zVDj · 2025-06-09
> >
> > The authors have submit a revision version and the revision solves part of my concerns. While the paper presents a complete study,  it offers only marginal progress in terms of the LLMs' reasoning capabilities.

---

> > > ### Author Response · Authors · 2025-06-09
> > >
> > > Dear Reviewer zVDj,
> > >
> > > Thank you for your time in re-evaluating our work and for providing your final thoughts.
> > >
> > > We appreciate your feedback on the revised manuscript. We are grateful for the constructive comments you provided throughout the review process.
> > >
> > > Sincerely,
> > >
> > > The Authors

---

### Review · Reviewer_poJi · 2025-06-03

**Summary Of Contributions:**

This paper introduces a framework called Cumulative Reasoning (CR) aimed at improving the reasoning capabilities of large language models (LLMs). The framework explicitly assigns the LLM three distinct roles throughout the reasoning process: proposer, verifier, and reporter. As the proposer, the LLM generates intermediate reasoning steps based on the given context. In the role of verifier, the LLM (or external reasoning systems) converts these intermediate steps into formal representations. Finally, as the reporter, the LLM synthesizes the final solution by integrating information from all intermediate steps. The design of this system is intuitive, and its effectiveness in enhancing the reasoning abilities of existing LLMs has been empirically demonstrated.

**Audience:**

Yes

**Broader Impact Concerns:**

No concern on the ethical implications of the work requires adding a Broader Impact Statement.

**Claims And Evidence:**

Yes

**Requested Changes:**

A major revision is required, as explained in my previous comments. In addition, the discussion of related work needs to be expanded to provide a more detailed comparison between the proposed framework and existing approaches in multi-agent reasoning enhancement, as well as self-critique methods in reasoning tasks.

**Strengths And Weaknesses:**

Strengths
- Comprehensive experiments are conducted across various base models and tasks, showcasing the framework's applicability.
- The implementation details of the framework are thoroughly explained, providing clarity and reproducibility.

Weaknesses
- The novelty of the proposed system is limited. The concept of utilizing LLMs for reasoning step generation, self-verification, and information integration to enhance reasoning performance has been extensively explored in existing literature. While the framework demonstrates performance improvements compared to traditional reasoning methods (e.g., CoT, ToT), its contribution is more technical in nature rather than a broader scientific advancement with wide-ranging impact on the research community.

Suggestions
- To enhance the novelty of this paper, I recommend that the authors delve deeper into the methodology and aim to uncover more general insights into the reasoning process. While the performance improvement is certainly an encouraging indication of the proposed framework's effectiveness, further exploration could focus on identifying the key components that drive its success and pinpointing the bottlenecks in other methods. Additionally, a comparison with other similar reasoning frameworks would provide valuable context and contribute to a more in-depth analysis. Beyond performance improvement, it would be important to include a quantitative evaluation and comparison of the computational overhead introduced by different reasoning methods, as this should be considered a critical metric.

---

> ### Author Response · Authors · 2025-06-04
> **Rebuttal Comment Part I**
>
> Dear reviewer, thank you sincerely for your thorough review and insightful feedback on our submission. We truly appreciate your positive remarks on our comprehensive experiments, the clarity of our implementation details, and the intuitive design of our CR framework. We are also very grateful for your constructive suggestions, which we believe will significantly strengthen our paper. Below, we try our best to address each of your points in detail.
>
> > Q1: The concept of utilizing LLMs for reasoning step generation, self-verification, and information integration to enhance reasoning ...
>
> A1: We respectfully acknowledge your perspective that individual components like step generation, verification, and integration have been explored in prior work. Indeed, our work builds upon this valuable body of research. However, we wish to clarify that the primary novelty of our Cumulative Reasoning (CR) framework lies not merely in the inclusion of these components, but in the specific, structured, and synergistic way we orchestrate them.
>
> 1. CR explicitly assigns three distinct roles (Proposer, Verifier, Reporter) to the LLM (or external tools for verification). We believe this structured decomposition, inspired by human cognitive processes and principles from intuitionistic logic and mathematical constructivism (as we discuss in Section 3), allows for a more controlled and robust reasoning process than previously shown.
>
> 2. A key differentiating factor, as highlighted in our Section 3.1 ("Comparison with CoT and ToT") and Figure 2, is CR's dynamic construction of a Directed Acyclic Graph (DAG) of validated reasoning steps. Unlike CoT's linear chains or ToT's tree exploration (which often prunes branches or explores a fixed breadth/depth), CR cumulatively stores and leverages all historically validated steps. This allows for more flexible and complex reasoning pathways, where subsequent steps can draw upon a growing, verified knowledge base specific to the problem instance. We argue this is a distinct mechanism not commonly found in methods that might perform verification but do not explicitly build and utilize such a cumulative graph for ongoing reasoning.
>
> 3.  We provide an argument (Theorem 1, Section 3.1) suggesting $P\_{\text{CoT-SC}} \leq P\_{\text{ToT}} \leq P\_{\text{CR}}$ under certain assumptions. This theoretical underpinning, combined with the significant empirical improvements we demonstrate (e.g., +24% in Game of 24, +43% relative on MATH Level 5 problems),  and strong performance on LogiQA, ProofWriter as detailed in Appendix B), suggests that the specific architecture of CR offers more than an incremental technical improvement.
>
> 4. We think that CR, through its structured approach and the dynamic interplay of its components, offers insights into how complex reasoning can be decomposed and systematically rebuilt. The success of CR across diverse tasks (logical inference, arithmetic reasoning, mathematical problem solving) points towards a more generalizable approach to enhancing LLM reasoning.
>
> We believe CR's design principles and the resulting performance gains contribute to the scientific understanding of how to elicit more robust and complex reasoning from LLMs. We will revise our introduction and discussion sections to more clearly articulate these aspects of CR's novelty and its broader scientific implications.

---

> ### Author Response · Authors · 2025-06-04
> **Rebuttal Comment Part II**
>
> > Q2: Delve deeper into the methodology and aim to uncover more general insights into the reasoning process. Identify the key components that drive its success and pinpointing the bottlenecks in other methods.
>
> A2: Thank you for this valuable suggestion. In Section 3, we introduce the Proposer, Verifier(s), and Reporter as key components. Our ablation study in Table 2 (table:ablation-folio) provides empirical evidence for the importance of the Verifier (e.g., CR accuracy drops from 73.03% to 64.23% on FOLIO-wiki with GPT-3.5-turbo when the Verifier is removed). The cumulative DAG is another core element, allowing CR to build upon verified intermediate steps, which we argue addresses error propagation issues in simpler CoT.
>
> In Section 3.1, we discuss how CoT can explore invalid branches due to its linear nature and lack of intermediate verification, and how ToT, while exploring multiple paths, might not leverage the full history of verified steps in the same cumulative manner as CR's DAG. Our conceptual experiment on the Game of 24 (Table 1) also illustrates how decomposition and intermediate verification (core to CR) improve accuracy compared to direct solving.
>
> We will expand Section 3 and our Conclusion to provide a more explicit discussion on: 1. The specific contribution of each role (Proposer, Verifier, Reporter) to the overall reasoning efficacy; 2. How the cumulative DAG structure specifically addresses limitations (e.g., error propagation, limited context utilization) of methods like CoT and ToT; 3. The general insights suggested by CR, such as the benefits of explicit role-based decomposition and iterative knowledge construction for complex reasoning.
>
> > Q3: A comparison with other similar reasoning frameworks would provide valuable context and contribute to a more in-depth analysis.
>
> A3: Thanks for your suggestion. While we compared CR with fundamental methods like CoT, CoT-SC, and ToT, we acknowledge the value of discussing CR in the context of other advanced reasoning frameworks.
>
> We will expand Section 5 ("Related Work") to include a more detailed discussion and comparison with other relevant frameworks that might share aspects like iterative refinement, verification, or multi-faceted approaches. This will include, but not be limited to, approaches that leverage self-critique or multi-agent interactions (as we detail further in point 4). We will clearly differentiate CR based on its specific Proposer-Verifier-Reporter architecture, the cumulative DAG mechanism for knowledge integration, and its demonstrated performance across a variety of demanding benchmarks.
>
> > Q4: Beyond performance improvement, it would be important to include a quantitative evaluation and comparison of the computational overhead introduced by different reasoning methods, as this should be considered a critical metric.
>
> A4: We agree that computational overhead is a critical metric. Firstly, please see existing metrics within our manuscript:
>
> 1. In addition to the metrics reported in the main paper (e.g., "# Visited States" for Game of 24 in Table 7 and "Iters" for MATH problems in Table 10), Appendix B provides further comparative results on computational overhead for datasets like LogiQA (Table 13), ProofWriter (Table 14), FOLIO-val (Table 15), and LD (Table 16).
>
> 2. These results consistently show that CR achieves competitive or superior accuracy while often utilizing a comparable or significantly lower number of visited states compared to methods like ToT. For instance, on LogiQA, CR achieves 45.25% accuracy with 17 visited states, while ToT reaches 43.02% with 19.87 states. On ProofWriter, CR achieves 71.67% with 16.76 states, outperforming ToT (70.33% with 24.57 states) and CoT-SC (69.33% with 16 states but lower accuracy). This demonstrates CR's efficiency in exploring the solution space.
>
> We will add a dedicated paragraph or subsection (likely in Section 4 or as part of the discussion for each experiment type) to explicitly discuss computational overhead, drawing upon the data from both the main paper and Appendix B. We will further elaborate on the trade-off: CR might incur more computational cost per step than simple CoT due to verification and multiple roles, but its improved accuracy and efficient state exploration (as shown across multiple datasets including those in Appendix B) often lead to a better overall balance of performance and resource usage, potentially being more efficient than exhaustive search or very wide ToT explorations for achieving similar high performance, that is, CR is more suitable for test-time scaling.

---

> ### Author Response · Authors · 2025-06-04
> **Rebuttal Comment III**
>
> > Q5: ...discussion of related work needs to be expanded to provide a more detailed comparison between the proposed framework and existing approaches in multi-agent reasoning enhancement, as well as self-critique methods in reasoning tasks.
>
> > A5: We agree this is crucial for positioning CR accurately. Our current Related Work section (Section 5) mentions works like Du et al. (2023) for multi-agent debate and several recent papers on self-criticism. We will substantially expand this:
>
> 1. Multi-agent Reasoning Enhancement: a) We will elaborate on how CR's Proposer-Verifier-Reporter roles can be viewed as a specialized multi-agent system. b) We will compare CR with other multi-agent approaches, highlighting differences in role definition (e.g., CR's predefined, functionally distinct roles vs. potentially more homogeneous agents or emergent roles in other systems), communication protocols (CR's information flow through the cumulative DAG), and coordination mechanisms. For instance, recent work by Wang et al. (arXiv:2503.18891, 2025) on dynamic agent elimination ("AgentDropout") explores adaptive team compositions, which contrasts with our fixed-role approach designed for specific reasoning structures. We will also discuss how CR differs from ensemble methods that simply aggregate outputs, by emphasizing the iterative and stateful collaboration between our defined roles.
>
> 2. Self-Critique Methods: a) We will discuss how CR's Verifier role aligns with and extends self-critique concepts. The Verifier can be an LLM performing self-critique or an external tool (like a code interpreter or symbolic system), offering a hybrid verification capability. b) We will differentiate CR by emphasizing its cumulative application of verification. While many self-critique methods refine a single reasoning trace or select among alternatives, CR uses verification to build a persistent, growing DAG of validated knowledge that informs all subsequent reasoning steps. This iterative accumulation and reuse of verified steps is a key distinction.
> c) We will connect this to works like Reflexion (Shinn et al., 2023) or Self-Refine (Madaan et al., 2023), pointing out similarities in the iterative improvement loop but also differences in the explicit role structure and the DAG-based knowledge accumulation in CR. Furthermore, emerging research such as that by Ke et al. (arXiv:2503.17363, 2025) on stepwise natural language self-critique ("PANEL") investigates verifiers that provide more nuanced feedback beyond binary validation, a direction CR could potentially incorporate in future work but currently differs from in its more discrete verification steps. We will also ensure to cite and discuss very recent advancements in verifier design and their integration into reasoning frameworks, such as the General-Reasoner framework (arXiv:2505.14652, 2025) which introduces a generative model-based verifier to enhance reasoning across various domains.
>
>
> Thank you again for your valuable time and constructive feedback.
>
> Sincerely,
>
> The authors

---

> > ### Comment · Reviewer_poJi · 2025-06-16
> > **Thanks for the clarifications and revisions**
> >
> > Thank you to the authors for the clarifications and revisions. However, my concerns persist. While constructing a DAG during the reasoning process and accumulating each reasoning step clearly improves reasoning accuracy and enhances efficiency in state exploration compared to the naive tree-of-thought search method, as demonstrated by the extensive experimental results, the overall contribution of this paper remains limited from my perspective. The proposed methodology is highly task-specific, the baselines are naive, and the ablation study results offer little in terms of new insights or informativeness.

---

> > > ### Author Response · Authors · 2025-06-16
> > >
> > > Dear Reviewer poJi,
> > >
> > > Thank you for your continued engagement with our manuscript and for providing this further clarification of your perspective. We sincerely appreciate the time you have invested. We understand your remaining concerns and would like to respectfully offer additional context on the contribution, generality, and empirical validation of the Cumulative Reasoning (CR) framework.
> > >
> > > ### On Contribution and Baselines
> > >
> > > We acknowledge your perspective on the contribution. Our methodology first establishes strong performance gains over widely-recognized foundational methods like Chain-of-Thought and Tree-of-Thought, as these are the standard benchmarks against which new reasoning architectures are commonly measured.
> > >
> > > However, a key part of our revision was to address this point by situating CR within the broader landscape of more advanced, contemporary reasoning frameworks. As detailed in the expanded Related Work section (Section 5), we now include comparisons with Graph-of-Thought (GoT), Forest-of-Thought (FoT), and various verifier-based approaches.
> > >
> > > A central element of CR is the Verifier role, which aligns with the growing body of research highlighting the critical importance of strong verifiers in complex reasoning pipelines. Our framework contributes a specific architecture for how verification can be integrated into a *cumulative* reasoning process, a direction also explored by recent work such as V-Star (Hosseini et al., COLM 2024). We believe that by providing a structured, DAG-based method for integrating verification, CR makes a solid contribution to this state-of-the-art research area.
> > >
> > > ### On the Generality of the Methodology
> > >
> > > We respectfully hold a different view on the framework's generality, and we believe our empirical results support this. A key strength of CR is its demonstrated versatility across three highly distinct and challenging domains:
> > >
> > > * **Logical Inference** (FOLIO-wiki, AutoTNLI), which requires formal reasoning and nuanced language understanding.
> > > * **Arithmetic Puzzle Solving** (Game of 24), a classic search and numerical reasoning task.
> > > * **Advanced Mathematics** (MATH dataset), which demands complex, multi-step symbolic reasoning across topics from algebra to number theory.
> > >
> > > The core CR architecture—orchestrating Proposer, Verifier, and Reporter roles to build a cumulative DAG—is consistent across all these tasks. The adaptation to each domain is achieved through role-specific prompts, a standard and necessary practice for applying any advanced LLM framework. The success of this single, consistent architecture across such varied problem types points to its robustness and general applicability.
> > >
> > > ### On the Insights from the Ablation Study
> > >
> > > We appreciate your feedback on the ablation study. Its primary purpose was to empirically validate our core architectural claims by isolating the impact of key components. We believe the results are highly informative for understanding *why* CR is effective.
> > >
> > > The finding that removing the Verifier causes a significant performance drop on the FOLIO-wiki dataset (from 73.03% to 64.23%) is a crucial piece of evidence. It demonstrates that the verification step is not superfluous but is a primary driver of CR’s improved accuracy. This confirms that actively identifying and filtering incorrect reasoning paths before they are integrated into the cumulative knowledge base is fundamental to the framework's success, offering a clear insight into its design principles.
> > >
> > > ***
> > >
> > > Thank you again for your critical perspective and your dedication to the review process. Your feedback has been instrumental in helping us strengthen the manuscript. We hope these clarifications, in light of the extensive revisions made, help to better articulate the novelty and significance of Cumulative Reasoning as a generalizable framework for enhancing complex problem-solving in LLMs.
> > >
> > > Sincerely,
> > >
> > > The Authors

---

### Author Response · Authors · 2025-06-04
**Summary of Revisions and Response to Reviews Part I**

Dear Editors and Esteemed Reviewers,

We sincerely thank you for your time, diligent effort, and invaluable feedback on our manuscript. Your insightful comments and constructive suggestions have been instrumental in helping us to significantly strengthen the paper.

We have carefully considered all the points raised and have undertaken a thorough revision of the manuscript. We believe these changes, guided by your feedback and our detailed rebuttals (provided previously on OpenReview: Rebuttal to Reviewer poJi - Comments I, II, III; Rebuttal to Reviewer zVDj - Comments I, II; Rebuttal to Reviewer vNxH - Comments I, II, and follow-up), address your concerns and enhance the clarity, rigor, and impact of our work.

Below, we summarize the key revisions made to the paper, followed by a more detailed outline of how we addressed the specific points raised by each reviewer.

**Summary of Major Revisions:**

1.  We have substantially revised the Introduction (Section 1) and Methodology (Section 3) to more clearly articulate the core novelty of Cumulative Reasoning (CR). This includes a greater emphasis on:
    * The structured, synergistic orchestration of the Proposer, Verifier(s), and Reporter roles.
    * The dynamic construction and utilization of a Directed Acyclic Graph (DAG) of validated reasoning steps, and how this cumulative, validated knowledge base enables more flexible and robust reasoning.
    * The philosophical underpinnings of CR (e.g., intuitionistic logic, mathematical constructivism) and their connection to CR's design.
    * A clearer distinction of CR's mechanisms from prior approaches like CoT and ToT (Section 3.1, Figure 3).
    * Refined definitions and explanations of key components like the Verifier, distinguishing between its conceptual role and practical implementations (LLM-based vs. symbolic/code-based, Section 3).

2.  Strengthened Comparison with Existing and Advanced Reasoning Frameworks:
    * Section 3.1 ("Comparison with CoT and ToT") has been significantly expanded to provide a more detailed structural and functional comparison, leveraging Figure 3 more effectively.
    * The Related Work section (Section 5) has been substantially updated and expanded to include detailed discussions and comparisons with a broader range of recent and advanced reasoning frameworks, including Graph-of-Thought (GoT), Self-Consistency with Voting (SCV)-like approaches, Forest of Thought (FoT), other proposer-verifier architectures, multi-agent reasoning systems (e.g., citing Wang et al., 2025 on AgentDropout), and various self-critique methodologies (e.g., citing Reflexion, Self-Refine, PANEL by Li et al. 2025, V\*-Reasoning by Hosseini et al. 2024, General-Reasoner 2025).

3.  In Section 3.1, we have re-evaluated and rephrased the discussion around Assumption 3.2 and Theorem 3.4. We now more clearly state the simplifying conditions (e.g., near-perfect verifier, unique correct path), their role in the illustrative theoretical model, and the limitations of these assumptions in general practical scenarios. We have also better connected this theoretical motivation to CR's empirical performance.

4.  Addressing concerns about computational overhead, we have introduced a new subsection (Section 3.2: "Computational Considerations"). This section discusses LLM interactions, verifier costs (LLM vs. symbolic), context management for the growing DAG, and the overall trade-off between computational investment and performance gains. We reference empirical data on iteration counts and visited states from our experiments (e.g., Tables 7, 10, and Appendix B tables) to support this discussion.

5.  Figure 2’s caption has been clarified to better represent the DAG dependencies in CR. Figure 3 is now more central to the comparison in Section 3.1.

6.  The entire manuscript has undergone revisions for clarity, precision, consistency (including chronological ordering of citations where appropriate, as in the "Self-Critique and Iterative Refinement" paragraph in Section 5), and overall rigor.

---

### Author Response · Authors · 2025-06-04
**Summary of Revisions and Response to Reviews Part II**

We would also like to briefly reiterate how the revisions address the main points from each review, building upon our individual rebuttals:

**To Reviewer poJi:**
We thank Reviewer poJi for the comprehensive review and constructive feedback.
1. As detailed in our rebuttal (A1) and implemented in the revised Introduction, Section 3, and Conclusion, we have significantly clarified CR's novelty, focusing on the structured orchestration of roles and the dynamic, cumulative DAG of verified knowledge. We have also better articulated the broader scientific implications.
2. The revisions to Section 3 (Methodology) and Section 3.1 now provide a more in-depth discussion on the contribution of each role, how the cumulative DAG addresses limitations of prior methods (e.g., error propagation, context utilization), and the general insights offered by CR's design, as promised in our rebuttal (A2).
3. Section 5 (Related Work) has been extensively expanded to provide detailed comparisons with other relevant advanced reasoning frameworks, differentiating CR based on its unique architecture and mechanisms, in line with our rebuttal (A3).
4. We have added a dedicated discussion in Section 3.2 ("Computational Considerations") analyzing computational costs and efficiency, drawing on empirical data from the main paper and Appendix B, as committed in our rebuttal (A4).
5. Section 5 now includes a substantially expanded discussion on multi-agent reasoning and self-critique methods, with detailed comparisons and citations to recent works as outlined in our rebuttal (A5).

**To Reviewer zVDj:**
We appreciate Reviewer zVDj's insightful comments and positive assessment of CR's core ideas.
1. In response to your feedback and as detailed in our rebuttal, we have revised Section 3.1 and Section 5 (Related Work) to explicitly discuss the conceptual relationships and key distinctions between CR and recent frameworks like GoT, SCV-like approaches, and FoT, highlighting CR's unique contributions.
2. As mentioned above (and in our rebuttal), we have added a comprehensive discussion on computational aspects in Section 3.2, contextualizing CR’s costs with its performance benefits and efficiency gains in search space exploration.

**To Reviewer vNxH:**
We are grateful to Reviewer vNxH for the very detailed and constructive feedback, which prompted significant clarifications.
1. Section 3 has been revised for greater clarity on role interactions, prompt design principles (with clearer guidance to Appendix F), and a more explicit connection of CR’s design to its philosophical inspirations (intuitionistic logic, mathematical constructivism), as discussed in our rebuttal (A1).
2. We have rephrased the Verifier's description in Section 3 to clearly distinguish its conceptual role from its practical implementations (LLM-based vs. symbolic/code-based), acknowledging the "near-perfect" verifier as an assumption for the theoretical model and emphasizing results with the code-based verifier, per our rebuttal (A2).
3. Section 3.1 has been significantly revised to better utilize Figure 3, detailing structural differences (CoT linearity, ToT tree, CR DAG) and elaborating on how CR’s DAG provides richer context. We also touch upon complexity trade-offs, as promised (A3). Figure 2's depiction of DAG dependencies has also been clarified in its caption.
4. We have carefully rephrased this section to clarify the simplifying nature of Assumption 3.2 for the theoretical model, better connecting it to empirical results and explicitly stating the limitations (e.g., unique path assumption). We also briefly acknowledge that LLM refinements are not always monotonic and the Verifier's role in mitigating this, as per our rebuttal (A4).
5. The Introduction, Discussion, new Section 3.2 (on computational considerations, referencing Appendix B and Table 7), and particularly Section 5 (Related Work) have been updated to more clearly articulate CR’s specific novelties (cumulative DAG, structured roles), expand comparisons, and include more recent literature, addressing your concerns (A5, A6).
We have clarified in the revised Figure 2 caption and text (Section 3) that the figure is illustrative and that CR aims to leverage a comprehensive context from previously validated propositions. We have also noted potential future work on optimizing DAG dependencies in the Conclusion.

We have endeavored to address all concerns comprehensively and believe the revised manuscript is now a much stronger contribution. We are hopeful that these revisions meet your expectations. We appreciate this opportunity to improve our work and are ready to provide any further clarifications if needed.

Thank you once again for your invaluable guidance.

Sincerely,

The authors

---

> ### Comment · Reviewer_vNxH · 2025-06-09
>
> The authors made a detailed revision of the paper, taking into account all my concerns, I very much appreciate it. I believe the paper drastically improved the first version, and now it delivers insights that can be interesting for the entire community. I still think that the theoretical formalisation is not very insightful, specifically, the assumption is too strong to hold in practice. However, I do understand why the authors want to keep it, and I am fine with it. The line of work is definitely interesting and worth to be explored more, I hope the authors will continue in that direction, possibly working with exact verifiers, that might be helpful to guarantee the assumption.

---

> > ### Author Response · Authors · 2025-06-09
> >
> > Dear Reviewer vNxH,
> >
> > Thank you very much for your positive and thoughtful response. We are delighted to hear that you found the revisions comprehensive and that you believe the paper has been significantly improved.
> >
> > Your suggestion to continue this line of work, particularly by exploring exact verifiers to better satisfy the assumption, is excellent and aligns perfectly with our vision for future research.
> >
> > Thank you once again for your invaluable feedback and guidance throughout this process. Your insights have been instrumental in strengthening our work.
> >
> > Sincerely,
> >
> > The Authors

---

> ### Comment · Reviewer_poJi · 2025-06-09
> **Highlighting Revisions in the Updated Version**
>
> Thank you to the authors for the clarification and revisions. It would be greatly appreciated if the authors could highlight the added or modified sections in the revised version compared to the original, as this would help the reviewers more easily identify the changes.

---

> > ### Author Response · Authors · 2025-06-10
> >
> > Dear Reviewer poJi,
> >
> > Thank you for your prompt and very helpful suggestion. We appreciate you pointing this out, as it will certainly facilitate the review process.
> >
> > In accordance with your request, we have now uploaded a new revision of our manuscript. In this version, all modified sections and additions have been highlighted in blue for easy identification.
> >
> > We hope this updated format is convenient for you and the other reviewers. Thank you again for your guidance.
> >
> > Sincerely,
> >
> > The Authors

---

### Decision · Action_Editor_x9sj · 2025-07-17

**Recommendation:** Accept as is

**Audience:**

Yes

**Audience Explanation:**

n/a

**Claims And Evidence:**

Yes

**Claims Explanation:**

The proposed cumulative reasoning method performs favourably wrt CoT, ToT and self-consistency baselines. As pointed by some reviewers the method is more costly than the above baselines as it must perform iterative verification and proposals. One of the limitations of the method is also that the fact that its prompts are heavily hand-engineered for every task, also pointed out by a reviewer. This to some extent plagues also a lot of other similar papers; I recommend acceptance of this paper due to the fact that it proposes a chain of thought method based on iterative verification and generation based on previously verified inferences, an idea that might spark interesting discussions and future methods.